# Characterizing trachoma elimination using serology

Trachoma is targeted for global elimination as a public health problem by 2030. Measurement of IgG antibodies in children is being considered for surveillance and programmatic decision-making. There are currently no programmatic guidelines based on serology, which represents a generalizable problem in seroepidemiology and disease elimination. Here, we collate *Chlamydia trachomatis* Pgp3 and CT694 IgG measurements from 48 serosurveys across Africa, Latin America, and the Pacific Islands (41,168 children ages 1–5 years) and propose a novel approach to estimate the probability that population *C. trachomatis* transmission is below or above levels requiring ongoing programmatic action. We determine that trachoma programs could halt control measures with >90% certainty when seroconversion rates (SCRs) are ≤2.2 per 100 person-years. Conversely, SCRs ≥4.5 per 100 person-years correspond with >90% certainty that further control interventions are needed. More extreme SCR thresholds correspond with higher levels of confidence of elimination (lower SCR) or ongoing action needed (higher SCR). This study demonstrates a robust approach for using trachoma serosurveys to guide elimination program decisions.

Trachoma, caused by repeated ocular infection with *Chlamydia trachomatis*, is targeted for global elimination as a public health problem (EPHP) by 2030[1,2]. The World Health Organization (WHO) defined EPHP based on clinical signs of trachoma, and significant progress has been made globally, with 18 countries validated to have achieved EPHP as of July 2024[3]. As countries approach and achieve EPHP, programs are considering the use of complementary measures of *C. trachomatis* infection to monitor population-level transmission[4–8]. Potential approaches include nucleic acid amplification-based tests, such as polymerase chain reaction (PCR), and serologic assays that measure immunoglobulin G (IgG) antibody responses in young children. Unlike PCR-detectable infection, which is transient, IgG responses provide a measure of previous infection that is sensitive as populations approach trachoma elimination. Previous studies of IgG responses to *C. trachomatis* have characterized Pgp3 and CT694 antigens as highly immunogenic[9]. Consistent shifts in population-level age-specific seroprevalence and seroconversion rates (SCR) to these antigens among children correspond with changes in prevalence of trachoma[6,10–12]. Multiplex IgG assays lend themselves to inexpensive, concurrent

surveillance of multiple diseases, including trachoma[13]. A key challenge remains: can surveys of serological responses reliably determine if *C. trachomatis* transmission falls below levels that require population-level trachoma interventions?

Deriving data-driven thresholds for intervention represents a generalizable problem for neglected tropical diseases and other infectious diseases, such as malaria. In the context of trachoma, some ocular *C. trachomatis* infections could still occur at very low levels of transmission, but cases of blindness from trachoma would be unlikely in the absence of repeated infections over many years[14]. Therefore, trachoma-specific population interventions in support of EPHP usually stop before interruption of ocular *C. trachomatis* transmission has been achieved. With this in mind, our focus was to characterize trachoma serology in relation to whether public health efforts were needed (or not).

Here, we combine data for IgG antibodies, PCR, and clinical observations from 63,911 children ages 1–9 years (41,168 ages 1–5 years) enrolled in 48 cross-sectional surveys across a gradient of trachoma prevalence settings to create a well-characterized trachoma

✉ e-mail: everlyn.kamau@ucsf.edu

serology dataset of unprecedented scale. Our objectives were to develop a serologic signature of trachoma elimination by examining the distribution of SCRs across many populations no longer requiring interventions against trachoma, to develop an approach that specifies thresholds of the SCR tied to programmatic action, and provide guidance for future surveys about the need for trachoma-specific interventions given an estimate of the SCR. The final step is akin to using a population-based serological survey as a diagnostic tool to obtain a post-test probability of whether the population should be treated – using a diagnostic testing paradigm at the population level[15]. The results provide new information to guide the use of serology to monitor trachoma as we approach the 2030 endgame and provide a generalizable example for how programs could advance data-driven thresholds for action based on specific biomarkers.

## Results

### Characterizing populations along a gradient of endemicity
The transition from endemic ocular *C. trachomatis* transmission causing blindness to interruption of transmission likely follows a continuum. We used the consensus of 10 expert reviewers and available clinical, PCR, and antibody data *(Methods)* to identify populations that fell at ends of the continuum corresponding with clear programmatic action: those at a high level of transmission that require additional trachoma-specific interventions to safeguard public health, and those that require no further interventions. Among the 48 study populations, 11 showed clear evidence of significant, ongoing transmission that required further intervention, 23 demonstrated clear evidence of trachoma control with no further program action needed, and 14 were unclassified (Table 1). Unclassified surveys were used to illustrate application of the methods and included (i) those for which there was no consensus regarding the need for intervention (five surveys in Ethiopia and Malawi), (ii) new baseline surveys and opportunistic serological surveys without PCR testing (five surveys in Sudan, Peru and Malaysia), and (iii) surveys in populations with unusual trachoma epidemiology where Trachomatous Inflammation−Follicular [TF] prevalence is above the EPHP threshold but Trachomatous Trichiasis [TT] prevalence is below the EPHP threshold, and biomarkers were inconsistent with clinical signs (four surveys in Papua New Guinea and Vanuatu). Each population was treated as an evaluation unit (EU), which is the normal administrative unit for health care management for trachoma interventions, typically representing 100,000 to 250,000 people[16]. Most EUs were surveyed following the Tropical Data protocol for trachoma, with villages as the primary sampling unit for cluster-based surveys[17], though five cluster randomized trials also contributed.

In the 34 classified EUs, age-specific Pgp3 seroprevalence flattened as populations approached and achieved trachoma control (Fig. 1), and SCRs decreased and approached 0 (Table 1). This initial result reinforced the previously established relationship between serology and other measures of *C. trachomatis* transmission[11]. Unclassified surveys represented a range of seroprevalence and seroconversion rates (Fig. 1, Supplementary Fig. 1).

### Seroconversion rate distributions by trachoma classification
We fitted a catalytic model that assumed a constant force of infection to estimate EU-level SCRs, which adequately fit the data given the narrow age range of 1–5 years *(Methods)*. SCR estimates were heterogenous across settings (Fig. 2A) but mostly separated between trachoma categories for the larger proportions of the combined posterior distributions (Fig. 2B). We focus on the SCR due to its epidemiologic interpretation, but the SCR was linearly related to seroprevalence (Supplementary Fig. 2), as previously shown in a narrower set of populations[11], and the overall pattern in SCR estimates across the gradient of transmission was similar when summarized as seroprevalence (Supplementary Fig. 3).

### Seroconversion rate thresholds to inform programmatic action
Clear biomarker thresholds can aid programmatic decision-making. We used SCR distributions estimated in 1–5-year-olds in the different EU categories to estimate a post-test probability that a population would fall in each category given an estimate of the SCR. We assumed a Bayesian mixture model to allow for transition between categories. For each category, we multiplied the combined EU-level SCR distributions in Fig. 2B (the likelihood) with a prior probability of each category, leading to a posterior probability of each category as a function of the SCR (Methods). We present main results for two sets of priors that reflect important potential use cases (Table 2). In near elimination settings, we assumed a moderately informative prior of 80% probability that a trachoma program could halt control measures, with the rationale that during or after post-treatment validation further treatments would be unlikely in the absence of recrudescence. In this scenario, the posterior probability that an EU would require no further action exceeds 90% when the SCR is ≤2.2 per 100 person-years (Fig. 3A). Conversely, SCR values ≥4.5 per 100 person-years correspond with >90% certainty that the population falls in the category of EUs in which further programmatic action is needed to control transmission. The choice of a particular threshold is ultimately a policy decision based on a specified level of confidence. More stringent (lower) SCR thresholds correspond with higher levels of confidence of elimination. For example, an SCR = 1.9 per 100 person-years corresponds with a level of confidence of 95% (Fig. 3A). In population surveys without strong prior information, such as baseline surveys or investigations with unusual epidemiology, an uninformative prior (50% probability of each category) may be more appropriate, with SCR value corresponding to 90% probability of no action needed equal to 1.6 per 100 person-years, down from 2.2 with an 80% prior (Fig. 3B, Supplementary Table 1). Notably, the estimated posterior probabilities were relatively insensitive to the assumed prior probabilities (Supplementary Fig. 4). The results were insensitive to the prior because there was a reasonably good separation in SCR distributions between the groups (Fig. 2). Additionally, the probability of elimination estimates were robust to exclusion of individual EUs and entire countries (Fig. 4). Data from Malawi and Ethiopia were most influential based on jackknife *n* − 1 posterior probability functions, but their influence was small in regions of the SCR near higher levels of confidence. That is, at ≥80% probability that no further action was needed, the difference between SCR when including all data and excluding either Malawi or Ethiopia was <0.6. For the analysis excluding individual EU as in Fig. 4, we show comparison of expert votes with the predicted category probabilities for each EU in Supplementary Table 2.

### Posterior probability of need for intervention in unclassified evaluation units
In future serological surveys, the methods proposed here lead to at least two useful probabilistic statements. First, pooled distributions of SCR in the different categories of endemicity (Fig. 2B) can be combined with prior probabilities of each category to obtain a posterior probability that a newly surveyed EU falls in each category. The approach treats a serological survey as a diagnostic test at the population level, akin to a laboratory assay at the individual patient level, leading to a post-test probability of programmatic action given the survey SCR estimate. Second, the probability that a population's SCR falls below a chosen threshold immediately follows from estimating the SCR and its uncertainty.

To illustrate how new surveys can be used to determine the need for programmatic action, or whether a population's SCR is below a specified threshold, we used the 14 EUs that were left unclassified. For each EU, we calculated the posterior probability of the need for additional programmatic action given its SCR distribution, assuming an informative prior probability of 80% that no programmatic action would be needed (Fig. 5A). Additionally, we determined the empirical

**Table 1 | Summary of trachomatous inflammation—follicular (TF), *Chlamydia trachomatis* infection prevalence, median Pgp3 IgG seroprevalence and mean seroconversion rate (SCR) by study evaluation unit (EU)**

| Trachoma Category / Survey (Country-EU-Year) | N children (1–5 years) | N clusters | TF prevalence (%) | Infection prevalence (%) | Seroprevalence (%) (95% CrI) | SCR per 100 person-years (95% CrI) |
|---|---|---|---|---|---|---|
| **Action needed** | | | | | | |
| Kiribati-Kiritimati-2016 | 219 | [c] | 30 | 26.8 | 39.3 (32.7–45.5) | 17.7 (13.6–22.4) |
| Kiribati-Tarawa-2016 | 615 | 22 | 41.5 | 29.1 | 50.8 (44.5–56.2) | 14.4 (10.6–19.2) |
| Ethiopia-WUHA/Wag Hemra-2016 | 4384 | 40 | 51 | 21.6 | 38.2 (32.5–44) | 13.4 (10.4–17.1) |
| Ethiopia-TAITU/Wag Hemra-2018 | 1487 | 48 | 54.3 | 16.7 | 33.3 (26.9–39.9) | 12.3 (9.4–15.8) |
| Ethiopia-Ebinat-2019 | 510 | 30 | 42.5 | 7.1 | 28.3 (21.2–35.7) | 9.4 (6.8–12.7) |
| Niger-PRET/Matameye-2013 | 1010 | 24 | 7.8 | 5.2 | 26.3 (18–34.2) | 9 (6.4–12.3) |
| Ethiopia-Andabet-2017 | 307 | 22 | 48 | 6.6 | 24.1 (15.2–33.3) | 7.7 (5.3–11) |
| United Republic of Tanzania-Kongwa-2013 | 2256 | 8 | 8.8 | 2.5 | 21.1 (11.1–31.2) | 5.4 (3.3–8.4) |
| Solomon Islands-Temotu/Rennel/Bellona-2015 | 259 | 13 | 14.3 | 1.8 | 16.4 (8.8–23.8) | 4.8 (2.9–7.6) |
| United Republic of Tanzania-Kongwa-2018 | 1307 | 50 | 7.1 | 3.5 | 12 (8.7–15.6) | 4.2 (3.1–5.6) |
| Ethiopia-Goncha-2019 | 344 | 30 | 16.7 | 1 | 7.3 (2–12.8) | 2.7 (1.6–4.3) |
| **Action not needed** | | | | | | |
| Malawi-Chapananga-2014 | 566 | 24 | 4.9 | 0.2 | 8.5 (5.3–11.8) | 2.9 (1.8–4.4) |
| Morocco-Agdaz-2019 | 578 | 30 | 0.2 | - | 7.6 (2.6–12.5) | 2.5 (1.6–3.8) |
| Malawi-Luzi Kochilira-2014 | 701 | 24 | 6.5 | 0.3 | 5.2 (3–7.6) | 1.8 (1.1–2.9) |
| Malawi-Kasisi/DHO-2014 | 599 | 24 | 4.7 | 0.2 | 4.5 (0.8–8.2) | 1.7 (1–2.7) |
| Malawi-DHO Nkwazi-2014 | 683 | 24 | 5.4 | 0.3 | 4.7 (2.9–6.6) | 1.6 (0.9–2.7) |
| Gambia-River Regions-2014 | 446 | 36 | 3.4 | - | 3.1 (1.2–5.1) | 1.1 (0.5–1.8) |
| Ghana-Wa-2016 | 835 | 24 | 1 | 0 | 2.8 (1.3–4.3) | 1 (0.5–1.7) |
| Ethiopia-Woreta Town-2021 | 427 | 30 | 2.9 | 0.8 | 2.5 (0.7–4.4) | 1 (0.5–1.7) |
| Ethiopia-Woreta Town-2017 | 166 | 12 | 2.7 | 0 | 2.4 (0–5.1) | 1 (0.3–2.4) |
| Ghana-Bole/Sawla-Tuna-Kalpa-2016 | 817 | 24 | 0.7 | 0.1 | 2.6 (1.2–4.1) | 0.9 (0.5–1.5) |
| Togo-Anie-2017 | 779 | 25 | 0.3 | - | 1.9 (0.8–3.1) | 0.7 (0.3–1.2) |
| Morocco-Boumalne Dades-2019 | 632 | 29 | 0 | - | 1.8 (0.6–3.1) | 0.6 (0.3–1.1) |
| Togo-Keran-2017 | 802 | 25 | 0.4 | - | 1.6 (0.1–3.2) | 0.6 (0.3–1) |
| Ethiopia-Metema-2021 | 497 | 30 | 3.2 | 0 | 1.6 (0.4–2.7) | 0.6 (0.3–1.1) |
| Ghana-Zabzugu Tatali-2016 | 845 | 23 | 1.3 | 0.2 | 1.8 (0.2–3.4) | 0.6 (0.3–1.1) |
| Ghana-Jirapa-2016 | 703 | 23 | 0.8 | [b] | 1.2 (0.2–2.3) | 0.4 (0.2–0.9) |
| Ghana-Nadowli-2016 | 787 | 24 | 0.9 | 0 | 1 (0.2–1.8) | 0.4 (0.1–0.8) |
| Ghana-West Gonja-2016 | 710 | 24 | 1.4 | 0 | 1.1 (0.1–2.1) | 0.4 (0.1–0.7) |
| Ghana-Gushegu Karagu-2016 | 845 | 24 | 0.9 | 0 | 1 (0.3–1.8) | 0.3 (0.1–0.7) |
| Ghana-Tolon Kumbugu-2016 | 916 | 24 | 1.1 | [b] | 0.9 (0.2–1.6) | 0.3 (0.1–0.7) |
| Ethiopia-Alefa-2017 | 316 | 22 | 3.2 | 0 | 0.6 (0–1.6) | 0.3 (0.1–0.7) |
| Ghana-Saboba Cherepen-2016 | 718 | 22 | 0.7 | [b] | 0.6 (0–1.1) | 0.2 (0.1–0.5) |
| Niger-MORDOR/Dosso-2015 | 5860 | 30 | 0.7 | 0 | 0.4 (0.1–0.6) | 0.1 (0–0.2) |
| **Unclassified** | | | | | | |
| Sudan-El Seraif-2019 | 749 | 30 | 13.7 | - | 25.4 (18.1–32.7) | 8.3 (6–11.3) |
| Sudan-Saraf Omrah-2019 | 697 | 35 | 10.9 | - | 23.5 (15.5–32.3) | 7.7 (5.5–10.5) |
| Ethiopia-Dera-2017 | 335 | 22 | 14.7 | 0 | 8.6 (1.4–14.6) | 3.1 (1.8–4.9) |
| Sudan-Kotom-2019 | 710 | 30 | 1.5 | - | 7.9 (4.1–11.9) | 2.6 (1.7–3.8) |
| Vanuatu-Torba/Malampa/Penama/Shefa/Tafea/Sanma-2016 | 634 | 33 | 16.5 | 1.8 | 7.7 (5.5–10) | 2.6 (1.6–3.8) |
| Malawi-Ngabu Ngokwe-2014 | 579 | 24 | 5.7 | 0.1 | 7.3 (3.3–11.4) | 2.5 (1.5–3.8) |
| Malawi-Mkanda Gumba-2014 | 694 | 24 | 7.2 | 0.6 | 6.9 (4.1–9.9) | 2.4 (1.5–3.7) |
| Papua New Guinea-Mendi-2015 [a] | 576 | [c] | 15.5 | 3.9 | 5.5 (3.6–7.4) | 2 (1.3–2.7) |
| Papua New Guinea-Daru-2015 [a] | 469 | 24 | 13.6 | 0 | 5.2 (2.9–7.7) | 1.8 (1–2.9) |

**Table 1 (continued) | Summary of trachomatous inflammation—follicular (TF), Chlamydia trachomatis infection prevalence, median Pgp3 IgG seroprevalence and mean seroconversion rate (SCR) by study evaluation unit (EU)**

| Trachoma Category / Survey (Country-EU-Year) Action needed | N children (1–5 years) | N clusters | TF prevalence (%) | Infection prevalence (%) | Seroprevalence (%) (95% CrI) | SCR per 100 person-years (95% CrI) |
|---|---|---|---|---|---|---|
| Ethiopia-Debay Tilatgin-2019 | 292 | 30 | 15.8 | 1.2 | 4.3 (1.2–7.2) | 1.6 (0.8–2.7) |
| Malaysia-Sabah-2015 | 1033 | 151 | - | - | 4.7 (3.3–6.1) | 1.5 (1–2) |
| Peru-Amazonia-2020 | 423 | 21 | - | - | 3.8 (1.7–5.8) | 1.3 (0.7–2.3) |
| Ethiopia-Machakel-2019 | 449 | 30 | 12.8 | 0.1 | 2.3 (0–4.5) | 0.9 (0.4–1.7) |
| Papua New Guinea-West New Britain-2015 | 602 | 27 | 12.8 | 2.4 | 1.5 (0.2–3) | 0.5 (0.2–1) |

*CrI* Bayesian credible interval, *PCR* polymerase chain reaction.

ᵃ In the Papua New Guinea surveys, *C. trachomatis* infection prevalence was measured by PCR only among children who had TF, not in all children as in the other included surveys.

ᵇ Three surveys from Ghana did not measure *C. trachomatis* infection prevalence by PCR but were classified alongside the other Ghana EUs as they were part of the same survey series that included 9 total EUs, 6 of which measured infection by PCR and were definitively considered as not requiring programmatic actions.

ᶜ In these surveys, each individual participant was treated as an independent observation (no cluster sample).

Surveys are grouped by trachoma category or programmatic decision (Methods) and ordered by SCR estimates as in Fig. 2 and Fig. 5. Here, seroprevalence and SCR were estimated for children aged 1–5 years, while TF and PCR-detected infection prevalence were estimated among children aged 1–9 years.

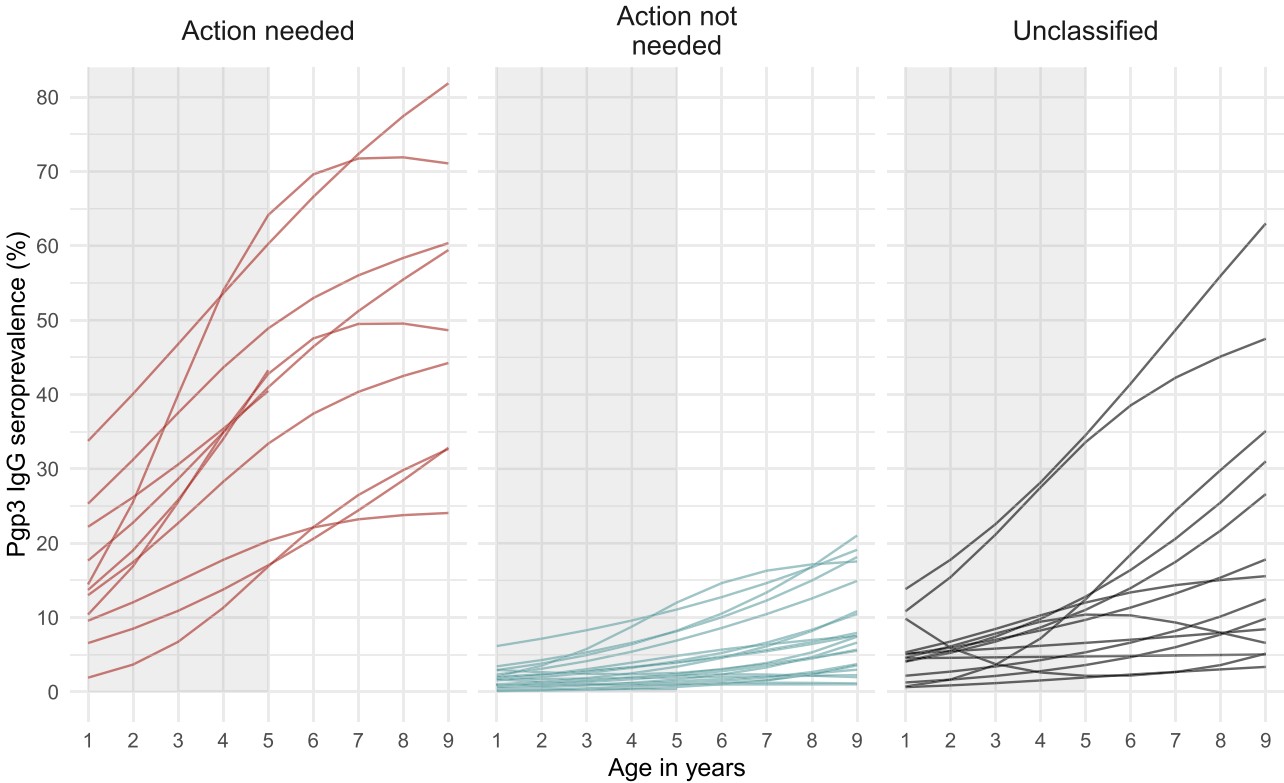

**Fig. 1 | Age-specific Pgp3 IgG seroprevalence among 1–9-year-olds.** Evaluation unit (EU)-level seroprevalence to *Chlamydia trachomatis* Pgp3 antigen among children aged 1–9 years (*N* = 48 evaluation units, and 63,911 children). Lines represent mean seroprevalence by age estimated using semiparametric cubic splines and EUs are grouped by categories based on programmatic responses (Methods). "Action needed" EUs include populations with clear evidence of ongoing transmission that require public health control measures, while "Action not needed" EUs include populations with demonstrated trachoma control. Unclassified EUs were used as a held-out sample in the analyses. The shaded region in each panel identifies the age range used in the main analyses: 1–5 years (41,168 children). Table 1 includes EU-specific sample sizes.

probability that an EU's SCR fell below an example threshold of 2.2 per 100 person-years (Fig. 5B, C), and compared the expert review votes of the unclassified EUs with the predicted category probabilities (Supplementary Table 3).

### Sensitivity analyses

Our main focus was characterizing Pgp3 serology in the age group 1–5 years, but we conducted sensitivity analyses that varied age ranges, single- vs dual-antigen testing, and catalytic model complexity. Owing to clear increases in seropositivity by age in all but the lowest transmission settings (Fig. 1), seroprevalence was generally lower if estimated in a narrower, younger age range compared with ages 1–9 years, but SCR estimates were consistent when estimated using different age ranges 1–3 years, 1–5 years, and 1–9 years (Supplementary Fig. 5). Seroprevalence and SCR estimates were lower if individual positivity required positive IgG responses to both Pgp3 and CT694 antigens,

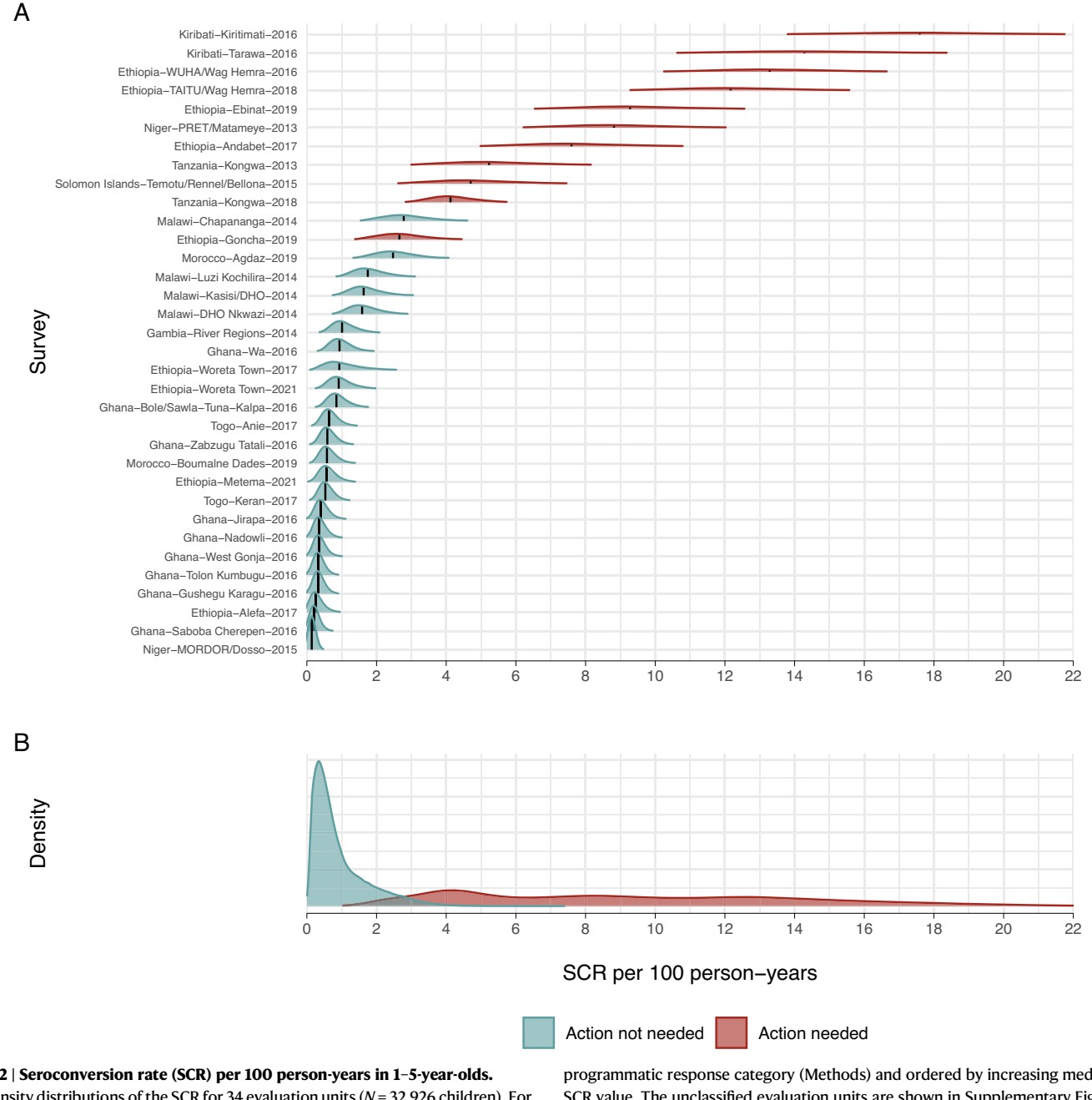

**Fig. 2 | Seroconversion rate (SCR) per 100 person-years in 1–5-year-olds.**
**A** Density distributions of the SCR for 34 evaluation units (*N* = 32,926 children). For each evaluation unit, the black vertical line shows the median estimate, and the density distributions depict the uncertainty about the median. EUs are colored by programmatic response category (Methods) and ordered by increasing median SCR value. The unclassified evaluation units are shown in Supplementary Fig. 1. **B** Pooled density distributions of the SCR for each category.

compared to requiring positivity to Pgp3 alone, but the magnitude of reductions was small (median difference 0.7% for seroprevalence and 0.3 per 100 person-years for SCR Supplementary Fig. 6).

Comparison of SCR estimates from the primary analysis with those from a reversible catalytic model allowing for seroreversion showed linear increases in the SCR due to model structure ($R^2 = 1$). As populations approach trachoma elimination, the differences in estimates are negligible, supporting a simplified model that ignores seroreversion (Supplementary Fig. 7). Finally, seroprevalence and SCR estimates from a generalized linear model aligned closely with Bayesian estimates (Supplementary Fig. 8). The Bayesian approach was a natural choice to generate parameter distributions (Fig. 2) and estimate posterior probabilities from a mixture model (Fig. 3), but comparability between estimates suggests that analysis of future monitoring surveys could use a simplified generalized linear modeling approach to estimate EU-level

seroprevalence and SCR. Source code available with this paper provides a didactic example of a simplified approach to estimating posterior probabilities for new surveys, as in Fig. 5.

### Generalization to alternative definitions of elimination

To illustrate how the approach could be used to develop SCR thresholds corresponding to interruption of ocular *C. trachomatis* transmission and generalize to applications with more than two population categories, we reclassified EUs using more stringent definitions based on PCR data and allowed for an intermediate category between extremes that included populations thought to be near interruption of transmission. Results were broadly consistent with the primary analysis focused on programmatic action, but with lower values of the SCR that correspond with a high level of certainty of being in the very low transmission group (Supplementary Information Text).

**Table 2 | Potential use of serological surveys to help inform programmatic response for three anticipated use cases and nine scenarios**

| Use case/scenario | Seroconversion rate | Programmatic Response/Examples |
|---|---|---|
| **Surveillance for elimination of trachoma after halting MDA or during post-validation (80% prior probability of No action needed)** | | |
| 1 | SCR ≤ 2.2 | No action needed. Morocco-Boumalne Dades-2019 (Fig. 2) |
| 2 | SCR > 2.2 & <4.5 | No action needed. Additional monitoring may be considered. Morocco-Agdaz-2019 (Fig. 2) |
| 3 | SCR ≥ 4.5 | Additional monitoring required (clinical, serology, PCR). |
| **Baseline survey to assess trachoma endemicity (50% prior probability of No action needed)** | | |
| 4 | SCR ≤ 1.6 | No action needed. Togo-Anie-2017, Togo-Keran-2017 (Fig. 2) |
| 5 | SCR > 1.6 & <3.8 | No action needed. Additional monitoring may be considered. Sudan-Kotom-2019 (Fig. 5) |
| 6 | SCR ≥ 3.8 | Consider initiating MDA. Sudan-El Seraif-2019, Sudan-Saraf Omrah-2019 (Fig. 5) |
| **Unusual epidemiology based on clinical and PCR markers (50% prior probability of No action needed)** | | |
| 7 | SCR ≤ 1.6 | Additional monitoring may be considered (serology, PCR) to assess etiology of clinical signs. Papua New Guinea-West New Britain-2015 (Table 1, Fig. 5) |
| 8 | SCR > 1.6 & <3.8 | Additional monitoring may be considered. Ethiopia-Dera-2017 (Table 1, Fig. 5) Vanuatu-Torba/Malampa/Penama/Shefa/Tafea/Sanma-2016 (Table 1, Fig. 5) |
| 9 | SCR ≥ 3.8 | Additional monitoring required (serology, PCR). |

The scenarios vary the seroconversion rate (SCR) estimated from Pgp3 IgG responses in children aged 1–5 years, per the primary analysis. Illustrative thresholds for the SCR have been provided as examples for how thresholds could be used to guide programmatic decision-making and were chosen for each use case using 90% posterior probability that action is not needed and 90% posterior probability that action is needed. In surveillance for elimination, illustrative thresholds reflect an informative prior assumption of 80% that no action is needed. For baseline survey and unusual epidemiology scenarios, illustrative thresholds reflect an uninformative prior (50% in each category). There were no examples in the present dataset of scenarios 3 and 9, but such results are possible.

## Discussion

Prevalence of TF has been instrumental in programmatic decision-making for trachoma over recent decades, and results from this study suggest that serology guidelines could provide a complementary tool as more populations approach and achieve EPHP. Characterizing the distribution of a key parameter, the SCR, across dozens of well-characterized populations enabled us to identify key regions of the SCR distribution that correspond with clear programmatic actions with specified levels of confidence. Beyond informing thresholds for stopping or resuming population-level interventions at a specified level of confidence, the method leads to another useful result. In the same way that clinicians estimate a post-test probability of disease based on a patient biomarker, we demonstrated how a population-level SCR distribution from a new serosurvey can be used to determine the population's post-test probability of a need for interventions, given distributions of the SCR from other well-characterized serosurveys. A similar analogy has been made between diagnostic tests and results from randomized controlled trials[15]. We also illustrated how future serosurveys estimate the probability that the population's SCR is below a defined threshold. Clear thresholds adopted by the community and endorsed by international organizations are easy to understand and can thus aid programmatic decision making. The probability that a population-level SCR is below a threshold combines both the magnitude of the SCR and its precision into a single number that is intuitive to decision makers.

How could the results be useful for programmatic decision-making? Serological surveys that demonstrate high probability of action needed (or not) will be most definitive, while those with SCRs in an intermediate range (e.g., >2.2 to <4.5 per 100 person-years) instead could lead to either additional measurements (e.g., PCR testing for infection) or consideration e.g., future monitoring depending on programmatic context. Below, we illustrate this general guidance through three different scenarios based on EUs that contributed to

these analyses (Table 2). First, in populations for which there is strong prior expectation of no action needed, such as having entered a period of post-treatment surveillance after halting antibiotic mass drug administration (MDA) or post-EPHP surveys, a population-level SCR below a defined threshold, would provide confirmatory evidence that no further population-level interventions are required. Ghana surveys provide examples of this scenario. In the same context, a survey that estimates a higher SCR could instead motivate additional inquiry. A second use is in baseline surveys where little is known about trachoma transmission and where serology can provide useful information in isolation or adjunct information to clinical signs. If serology suggests high probability of no action being needed, then programs could be confident in not initiating control activities or further investigation; Togo surveys illustrate this use case. Finally, serology can provide an objective characterization of *C. trachomatis* transmission in populations with persistent or recrudescent trachoma, or unusual epidemiology such as those characterized by high TF prevalence estimates but low prevalence of PCR-detected infection. Populations in Papua New Guinea and Vanuatu with TF prevalence 12–16% yet low SCRs are good examples of unusual epidemiology (Table 1). In these examples, SCR estimates are consistent with a high probability of no action being needed (Fig. 5). Additional serology surveys and monitoring for PCR-detected infection could help support program decision-making, such as whether MDA would be justified or whether there is a potentially different etiologic cause of TF.

The path to interruption of *C. trachomatis* transmission likely follows a continuous gradient in SCR, which makes specifying a single threshold to guide programmatic decision difficult and represents a broader challenge beyond trachoma. The approach developed here allows for this complexity and represents a methodologic advance in the use of serology to inform data-driven, programmatic guidelines. Using clinical and PCR measures of ocular *C. trachomatis* infection to identify populations that fell into clear categories of programmatic

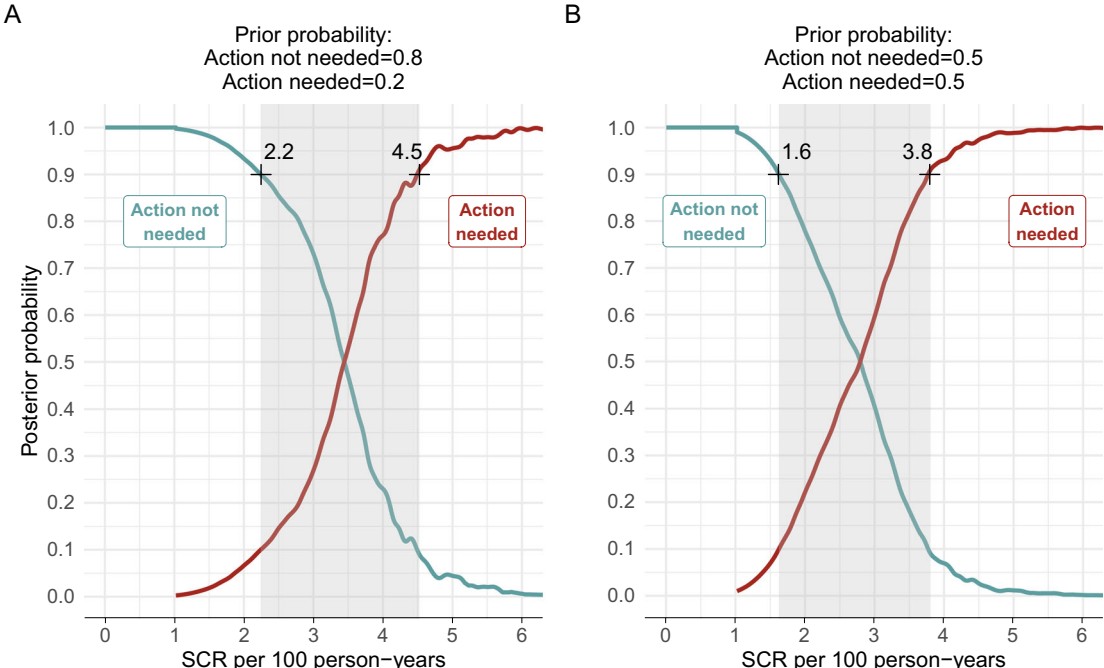

**Fig. 3 | Posterior probability of the need for population-level trachoma interventions using seroconversion rate.** Posterior probability of programmatic 'Action not needed' versus 'Action needed' categories along a range of seroconversion rates (SCRs) among 1–5-year-olds calculated using a two-component Bayesian mixture model (Methods). **A** Posterior functions assume moderately informative prior probabilities of 80% 'Action not needed' and 20% 'Action needed'. In principle, the posterior probability functions allow for the selection of thresholds to inform decisions based on serological surveys with a desired level of certainty.

For example, at a ≥ 90% level of certainty, SCR of ≤2.2 per 100 person-years corresponds to a posterior probability of 'Action not needed' and a SCR of ≥4.5 corresponds to a posterior probability of 'Action needed'. SCR values > 2.2 and <4.5 per 100 person-years may require additional information to inform programmatic action. **B** Posterior functions assume an uninformative prior of 50% 'Action not needed' and 50% 'Action needed'. Sensitivity analyses in Supplementary Fig. 4 demonstrate that posterior probabilities are insensitive to the prior assumptions.

decision-making, we developed a statistical approach that leads to probabilistic statements of whether further programmatic action is needed. The result is that decision makers can identify values of SCR that correspond with a specified level of certainty, for example ≥90% probability of no action being needed corresponding with SCR values ≤ 2.2 per 100 person-years (Fig. 3). Intuitively, higher levels of confidence lead to lower, more stringent SCR thresholds. Increasing the level of confidence to 95% corresponds with an SCR ≤ 1.9 per 100 person-years.

Standard classification techniques, such as a receiver operator characteristic curve, provide an alternative approach to identifying thresholds from a continuous measure. In these data, the SCR was an almost perfect classifier of programmatic action (Fig. 2). A cutoff in the SCR of 2.6 per 100 person-years (area under the curve = 0.99) that optimizes sensitivity and specificity (the Youden's J statistic) corresponds with the SCR value where posterior probability curves cross under an uninformative prior (Fig. 3B). This link illustrates how the Bayesian mixture approach enables additional information to inform thresholds through a prior probability of whether action is needed, and the certainty required to start or stop a program — effectively shifting a threshold to be more or less conservative depending on expectations and level of confidence desired.

This study extends earlier efforts to inform decision-making thresholds using serology by using data from more diverse populations and by advancing the methodology. Yet, this led to results broadly consistent with previous estimates based on alternative methods, suggesting robustness in the overall area of research. In a subset of EUs studied here, previous analysis classified individual sampling clusters based on PCR-detected infection status and found that an SCR ≤ 2.75 per 100 person-years had 90% sensitivity to identify clusters with any ocular *C. trachomatis* infection (AUC = 0.91)[11].

Another previous effort regressed population-level SCR values against TF prevalence and estimated that the current TF < 5% threshold for EPHP corresponded with a SCR of 1.5 per 100 person-years (95% CI: 0.0–4.9)[10]. In the United Republic of Tanzania, a population with 5.2% seroprevalence among children ages 1–3 years showed no evidence of trachoma re-emergence four years after cessation of antibiotic MDA[18]. Using linear mapping between seroprevalence and SCR (Supplementary Fig. 2), 5.2% seroprevalence corresponds to a SCR of 1.9 per 100 person-years. Diverse approaches to analysis and inference thus all converge on a narrow region of the SCR (1.5 to 2.8 per 100 child years) to delineate the threshold under different definitions. Moving forward, the present approach has advantages over previous efforts because it aligns with the current spatial scale of programmatic decision making (EUs), delineates EU categories using a process of expert consensus, and leads to a posterior probability of whether further control measures are needed as a continuous function of the SCR, allowing stakeholders to draft guidelines based on a specified level of confidence. Furthermore, it naturally accommodates new data from future surveys, which could then update the pooled SCR distributions and subsequent posterior probability estimates.

The analysis focused on the Pgp3 SCR among 1–5-year-olds, which was one of many variations across single- versus dual-antigen, age ranges, and population parameters (SCR versus seroprevalence that we evaluated. The addition of a second antigen, CT694 to the estimates could potentially improve specificity but did not dramatically reduce seroprevalence or the SCR, particularly near EPHP. A focus on Pgp3 alone should be sufficient given the added complexity of dual-antigen testing, particularly in the context of rapid diagnostic tests. A caveat is that most surveys measured IgG on the Luminex platform (Supplementary Table 1). Results should be comparable but not perfectly equivalent with other platforms, and there is always a possibility

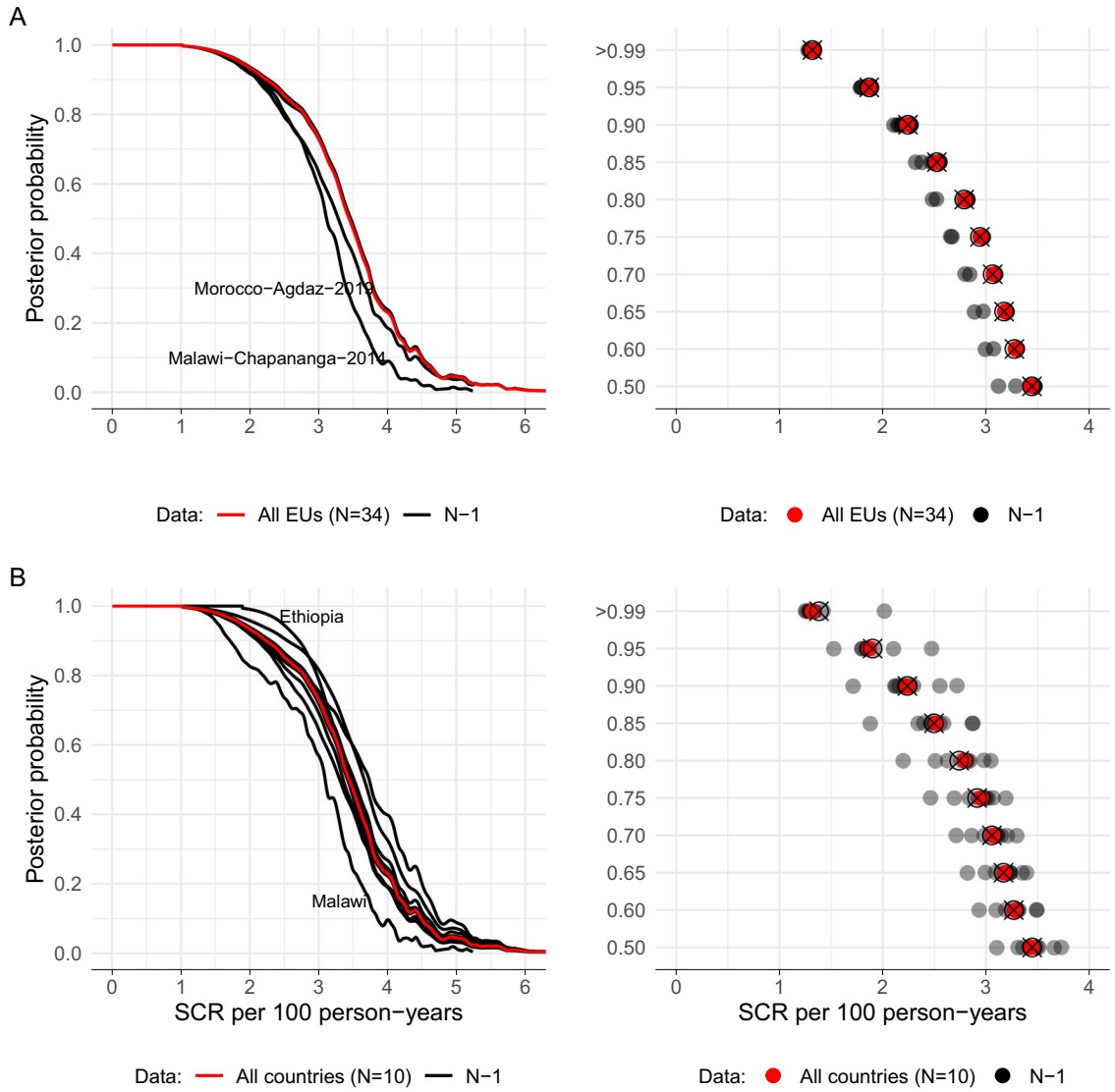

**Fig. 4 | Sensitivity analysis of exclusion of evaluation unit- and country-level data.** A jackknife $n-1$ resampling approach was used to iteratively alter group composition in a Bayesian Mixture model (Methods). **A** Posterior probability of No Action Needed for trachoma control removing each of 34 evaluation units (EUs) in turn. The red line summarizes the curve fit to all the data, and dark lines show $n$ jackknife subsample fits. In the right panel, red points mark the Seroconversion Rate (SCR) that corresponds with specific posterior probabilities of No Action Needed (in the right panel the Y-axis is zoomed in to the region of 0.5 to 1 to better display estimates). Gray points mark leave-one-out replicates, and open circles with a 'x' symbol indicate the mean SCR over the $n$ jackknife subsamples. Differences between full data estimates (red points) and open circles with a 'x' provide a jackknife estimate of bias, demonstrating no evidence of bias. **B** Posterior probability curves and SCR estimates that correspond with specific posterior probabilities as in (**A**), but with all EUs from entire countries left out of each jackknife replicate ($N = 10$ countries). All estimates assumed an 80% prior probability of no action needed. The two most influential held-out units are labeled in each sensitivity analysis. Overall, there was minimal effect of removing data at EU- or country-level in the higher posterior probabilities (>0.8) – our primary focus. More so, there was an overlap of posterior probabilities and corresponding SCR values of the reduced datasets with that of the original full sample.

of false positives or exposure to non-ocular *C. trachomatis* infections[6,19,20]. The 1–5 years age range is narrower than the current 1–9 years standard for TF surveillance but has practical advantages: it facilitates relatively reproducible household surveys, since these children are preschool aged, and IgG detected will reflect infections only in the preceding 6 years. The 1–3-year-old age range would provide a narrower infection history based on IgG, which may be ideal, but in many settings, it will be difficult to identify enough 1–3-year-olds per sampled cluster to assure survey rigor. Finally, although there was linear mapping between seroprevalence and SCR at EU level (Supplementary Fig. 2), the SCR should be preferable to guide decision-making because it implicitly adjusts for age, while seroprevalence estimates will differ when estimated in different age ranges due to increasing age-seroprevalence in settings with ongoing transmission

(Supplementary Fig. 5). Seroprevalence could also be influenced by exposure at birth to maternal urogenital *C. trachomatis* infection[6,21,22], yet seroprevalence would not increase with increasing age in the absence of ocular transmission to children. In these circumstances, population-level SCR should remain close to zero even with a higher seroprevalence in 1-year-olds.

This study had limitations. First, the process used to categorize EUs into groups that required public health action or not was based on clinical signs, PCR and serology data and involved an iterative process among the investigator team that ultimately relied on judgement. Separation of SCR distributions between categories was evident (Fig. 2), and estimates were insensitive to excluding individual surveys or countries (Fig. 4), but alternative approaches to defining categories could result in different SCR thresholds. In the Supplementary

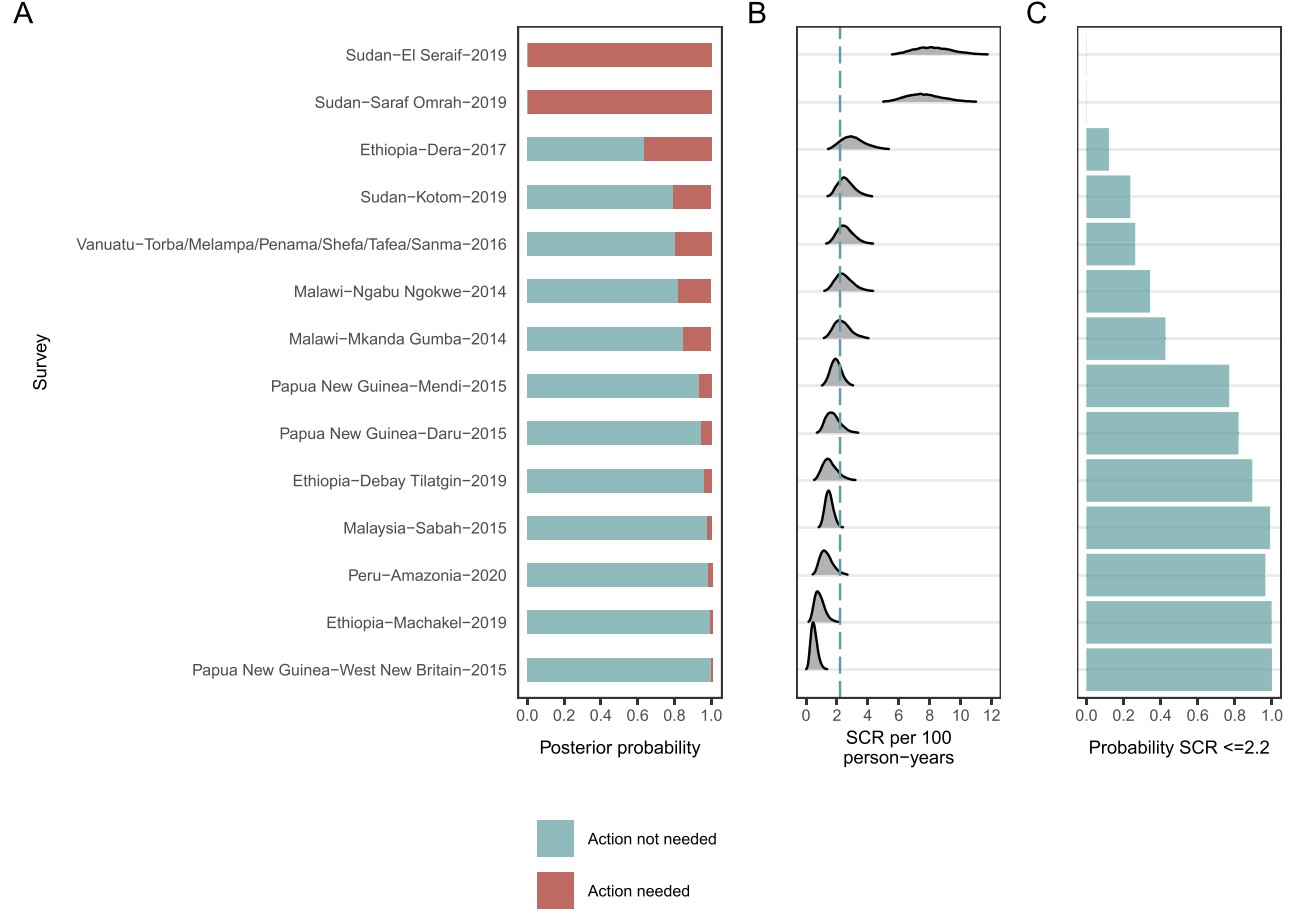

**Fig. 5 | Posterior probability estimates for unclassified evaluation units (EUs).**
**A** Probability of need or no need for trachoma program intervention in unclassified EUs. Unclassified EUs included baseline surveys in new populations that did not have PCR data (Sudan, Peru), opportunistic surveys not focused on trachoma (Malaysia), settings with unusual epidemiology based on trachoma biomarkers (Papua New Guinea, Vanuatu), and those that failed to achieve a consensus classification into 'Action not needed' and 'Action needed' categories (five from Ethiopia and Malawi). The posterior probability was calculated using seroconversion rate (SCR) estimates among 1–5-year-olds in a Bayesian mixture model that assumed prior probabilities of 80% for 'Action not needed' and 20% for 'Action needed'. EUs are ordered by increasing median SCR value shown in (**B**). **B** EU-specific SCR density distributions, with an example threshold shown at 2.2 per 100 person-years. **C** An illustrative threshold of 2.2 per 100 person-years corresponding to the 90% posterior probability ('+' in Fig. 3) was used to calculate the empirical probability of 'Action not needed' as the proportion of the SCR density distribution ≤2.2. Table 1 includes additional details for the unclassified EU populations.

Information Text, we provide an example of categorizing EUs based on PCR but without serology data that provides more stringent posterior probability thresholds. Second, we used a sample of serologic surveys primarily from the research context, which may have over-emphasized EPHP settings and ambiguous transmission scenarios. We addressed this by grouping EUs into categories based on clear programmatic action then fitting the SCR estimates to probability functions, while retaining EUs without clear category membership as an unclassified sample. The 34 EUs used to fit the probability functions represented a continuous gradient of the SCR (Fig. 3), which suggests the analysis adequately captured the transition from endemic to interruption of transmission. As increasing number of routine trachoma surveys incorporate the collection of serology data, further data will be available from a wider range of epidemiological contexts to validate and, if necessary, refine the approach presented in this paper. Third, after we found a more complex model with seroreversion led to relatively small increases in SCR estimates (Supplementary Fig. 7), we assumed a simple model to estimate the SCR that ignored seroreversion. Models that ignore seroreversion are easier to fit using standard regression techniques, but it means that estimates of SCR in future surveys would need to use the same underlying model and age range if comparing

estimates against thresholds developed here. Finally, SCRs were estimated from cluster surveys typically optimized for TF among 1–9-year-olds. In general, there were sufficient data at the cluster- and EU-level for valid analyses among 1–5-year-olds, but an important area of future work will be to develop guidance for cluster survey designs optimized to estimate EU-level seroprevalence and SCR, preferably using comparable antibody testing platforms.

Beyond advances in survey design for trachoma serology, this study prompts additional areas of future work. One, the posterior probability functions estimated here could potentially inform guidelines that specify SCR thresholds used to stop or start population-level trachoma control programs based on a specified level of confidence, as determined by programmatic stakeholders. Two, the probability below threshold estimates presented here (Fig. 5C) were inspired by a geostatistical modeling framework, and so a natural extension may be to use geospatial design and analysis for the SCR, considering cluster locations and within-EU heterogeneity[23,24]. Three, we lacked sufficient data to study whether repeated surveys in a single EU provide opportunities to assess the predictive value of posterior probability estimates. Several populations were measured repeatedly over time − Wag Hemra and Woreta Town in Ethiopia and Dosso, Niger. (In Dosso,

Niger, all data were combined into a single estimate, Niger-MORDOR/ Dosso, but seroprevalence was ≤0.6% in every survey[25]). The consistently low SCRs in Woreta Town and Dosso provide some proof of concept for the approach, but repeated surveys separated by multiple years in locations with moderate probability of no action needed would lead to higher posterior probabilities of no action needed or, potentially, detect recrudescence. Four, populations included in the analysis reflected a broad range of conditions and timing with respect to MDA treatment. Assessing whether timing between MDA and a serosurvey influences SCR estimates could be an area of future research. Finally, we identified levels of the SCR that correspond with trachoma program actions, but it remains unknown how current markers of infection in childhood (TF, PCR, serology) relate to future incidence of trichiasis and blindness from trachoma. The dynamic nature of transmission and the long timescale required to develop these complications make empirical measurements difficult, but modeling approaches could help fill the gap.

WHO guidance based on prevalence of TF has been central to the success of the global trachoma elimination program and we show here that a data-driven guideline based on serology could play a complementary role as we approach the trachoma endgame. Synthesis of extensive clinical, PCR, and antibody data enabled characterization of Pgp3 IgG in settings where population-level intervention was (or was not) clearly needed, and represents a new opportunity to develop an approach for programmatic decision-making based on a population's SCR. The approach represents a generalizable example for how to develop data-driven thresholds of elimination and for how serological surveys could be used to inform disease elimination programs.

## Methods

### Study sites and data sources
We gathered population-based serology data conducted in 48 surveys across EUs in 15 countries: Ethiopia ($n = 12$), The Gambia ($n = 1$), Ghana ($n = 9$), Kiribati ($n = 2$), Malawi ($n = 6$), Morocco ($n = 2$), Malaysia ($n = 1$), Niger ($n = 2$), Papua New Guinea ($n = 3$), Peru ($n = 1$), Solomon Islands ($n = 1$), Sudan ($n = 3$), Togo ($n = 2$), United Republic of Tanzania ($n = 2$) and Vanuatu ($n = 1$). The data were from published trachoma serology surveys with an emphasis on IgG antibody responses to Pgp3 collected among children ages 1–9 years and relatively recent reports. An EU is defined by WHO for trachoma control purposes as the administrative unit in which trachoma activities take place, typically consisting of 100,000–250,000 people[16]. Each EU included 20–30 clusters, where a group of households − typically in a single village − defined a study cluster. All surveys were conducted between 2013 and 2021, and demographic information on individual's age and household membership was collected. The sampled population comprised children ages <10 years since trachoma control programs currently make MDA decisions on the basis of the prevalence of TF in children ages 1–9 years[16]. Full descriptions of survey design, sampling units and geographical areas for the 48 surveys were previously published and summarized in Supplementary Table 4. The surveys included anti-Pgp3 IgG antibody measurements alongside clinical measurements in standard monitoring surveys and a small number of clinical trials. All surveys used population-based random and/or quasi-random sampling. Besides obtaining serology results for each survey, we also obtained individual- and population-level data on TF and PCR for ocular *C. trachomatis* infection, if available. (Supplementary Information Text includes detailed descriptions of clinical and specimen testing). In total, there were 63,911 individual observations from 1 to 9-year-olds, 41,168 from 1 to 5-year-olds, and 24,353 from 1 to 3-year-olds. Our principal focus was on anti-Pgp3 IgG antibody responses, but supplementary analyses included results based on a dual antigen approach, Pgp3 and CT694. In all data, age was measured in years.

### Classification of surveys based on trachoma program action
Progression to interruption of transmission is likely a continuum but, as we detail below, making probabilistic statements about whether an EU has reached a sufficiently low level of ocular *C. trachomatis* infection that population-level interventions against trachoma could stop, would be valuable for program decision-making. An initial summary of serology estimates by EUs demonstrated a continuous gradient in the distribution of seroprevalence and SCR values from high to low trachoma endemicity, with no natural "breakpoint". We identified populations at either ends of the gradient congruent with programmatic responses: (i) 'action needed', those with clearly high endemicity likely to lead to development of disease sequalae and blindness from trachoma in the absence of interventions, and (ii) 'action not needed', those with very low levels of infection with exceedingly small possibility of sufficient and sustained ocular transmission leading to blindness from trachoma, and thus no justification for population-level interventions, such as antibiotic MDA. Identifying the two domains that correspond with clear programmatic action allowed for the possibility that some populations would fall between the two extremes as they are in transition or have unusual epidemiology, and therefore further inquiry is needed, or a 'wait and watch' approach could be adopted, dependent on context[26,27].

We used an expert assessment of 10 coauthors with a range of knowledge of trachoma epidemiology and programmatic activities in each country to independently group EUs into one of the two categories, based on the above category descriptions and all available information, including summaries of clinical signs (TF, trachomatous inflammation−intense [TI]), PCR and serology. Raters could leave an EU unclassified if they felt it was unclear whether further trachoma-specific interventions would be needed or not (a copy of the dossier provided to raters and the rating results are provided in the repository, https://osf.io/va8uc/). EUs with ≥7/10 agreement on the category between raters were considered a consensus classification. EUs without a consensus classification (five from Ethiopia [$n = 3$] and Malawi [$n = 2$]) were left unclassified and were retained in the unclassified sample. The unclassified sample additionally included new baseline surveys and opportunistic serological surveys without PCR testing (five from Sudan [$n = 3$], Peru [$n = 1$]), and Malaysia [$n = 1$]), and surveys in populations with unusual epidemiology for trachoma based on available biomarkers (three EUs from Papua New Guinea and one survey from Vanuatu).

### Age-specific seroprevalence estimation
We used semi-quantitative IgG antibody responses to the Pgp3 antigen to identify samples that were seropositive and seronegative using survey-specific receiver operating curve (ROC)-derived cutoffs based on known positive and negative control samples with high sensitivity and specificity for most surveys, and a finite mixture model in the case of the Malawi and Malaysia surveys (Supplementary Information Text). We estimated seroprevalence by age using semiparametric cubic splines in a generalized additive model to allow for potential non-linear relationships with age, specifying binomial errors for seroprevalence, and random effects for clusters to account for repeated observations[28]. Seroprevalence increased with age at higher levels of transmission, but seroprevalence estimates throughout the paper were not age-adjusted as the adjustment made little difference over the narrow age ranges considered (Supplementary Fig. 9).

### EU-level seroprevalence and seroconversion rate estimation
We estimated seroprevalence and SCR, $\lambda$, as the two main serology-based summary measures. SCR is a serological measure for the force of infection (FOI), the rate at which susceptible individuals acquire infection.

EU-level seroprevalence estimates were calculated using a Bayesian extension of a generalized linear mixed effects model with a

random intercept per sampling cluster,

$$(seroprev \sim 1 + (1|cluster) + \varepsilon)\,(family = \text{gaussian}) \quad (1)$$

where the model response variable was antibody presence given as a binary variable (0,1). The models for seroprevalence estimation were implemented within the R package *rstanarm*[29] using weakly informative priors, $N(0, 10)$, for model parameters.

We estimated SCR in a catalytic model, where the probability of being seropositive as a function of age, $P_a$, or the proportion seropositive at age $a$, is given by,

$$P_a = 1 - e^{-\lambda a} \quad (2)$$

modeled in a binomial likelihood as $z_a \sim B(N_a, P_a)$, where $z$ is the number of seropositive individuals and $N$ is the sample size. We assumed a constant SCR over the age range, as previous analyses of 14 studies in this dataset demonstrated that a model with constant SCR fit the data as well as an age-varying SCR[11]. In a hierarchical structure, each cluster $j$ had a different SCR drawn from a common distribution,

$$\lambda_j \sim \exp\left(\lambda^{-1}\right) \quad (3)$$

where the hyper-prior $\lambda$ is a shared random variable representing the overarching EU-level SCR parameter fitted from data using an exponential prior distribution, $\lambda \sim \exp(1)$, which is a suitable prior to model a constant rate of infection events in a year. We fitted the catalytic models ignoring IgG waning to the seroprevalence data using *Stan* in R, using a Monte Carlo Markov Chain (MCMC) approach[30].

## Estimating the posterior probability of category

To make probability statements for each category, we used a mixture model framework applied to the combined full posterior distributions of EU-level SCR estimates. This approach assumed that each SCR estimate is drawn independently from a 2-component distribution of the two categories defined above, $k \in \{1, 2\}$. So, for each category or component ($C_k$ : action not needed, action needed) and SCR estimate, $x\varepsilon\mathbb{R}$, we computed the posterior probability, $p(C_k|x)$, using Bayes' rule:

$$p(C_k|x) \propto p(C_k) * \frac{p(x|C_k)}{p(x)} \quad (4)$$

where $p(C_k)$ is the prior probability that a population is in category $C_k$; and $p(x|C)$, is the likelihood evaluated as empirical probability density function at each MCMC draw $x$. $p(x)$ denotes the marginal likelihood or normalizing constant for the posterior density obtained by integrating the products of the likelihood, $p(x|C_k)$, and the prior probability. That is, the sum of the products of the density function and prior probability for each $k$,

$$\sum_{k=1}^{2} \omega_k f_k(x|C_k) \quad (5)$$

The prior probabilities were defined such that they sum up to one, i.e., $\sum_k \omega_k = 1$. The expression $p(x, |, C)/p(x)$ forms the likelihood ratio in the Bayesian mixture model. In a sensitivity analysis, we compared five sets of prior probabilities, or mixture weights, with increasing weight of $p(C_{k=\text{Action not needed}}) = \{0.5, 0.65, 0.70, 0.75, 0.80\}$ reflecting scenarios where there may be more prior certainty that no action is needed. The prior probabilities of the 'action needed' were computed as the complement value: $1 - p(C_{k=\text{Action not needed}})$.

Of the 48 EUs, 34 could be classified into the two categories. For the remaining 14 unclassified EUs, we used the above mixture model approach to estimate the EUs' posterior probability of being in each category.

## Serologic thresholds for programmatic action

We plotted the estimated posterior probabilities, $p(C_k|x)$, against the SCR ($x$) for each category and used the probabilities to identify example thresholds at which there is high posterior probability of being in each category. For both categories, we identified regions of the SCR where the $p(C_k|x) \geq 90\%$, corresponding to a high level of confidence in a program's need to deliver trachoma interventions (action needed) or not (action not needed).

To assess the robustness of estimates to inclusion of individual EUs or countries, we used an $n-1$ jackknife resampling approach to re-estimate the posterior probability values[31]. Given the full classified dataset of $[n_{eu}] = 34$ EUs from $[n_{country}] = 10$ countries, we repeated estimation of posterior probability of being in the 'Action not needed' category for each subsample $[i] = \{1, .., n\}$ of size $n-1$ obtained by leaving out one EU or country iteratively. We then aggregated the SCR values $[c]$ at each posterior probability $[p] = \{0.5, 0.6, 0.65, 0.7, 0.75, 0.8, 0.85, 0.9, 0.95, 0.99\}$ across the $n$ subsamples, computed their mean, and compared the means with the SCR values determined at the same $[p]$ probabilities using the full dataset as a jackknife estimate of the bias. In each jackknife $n-1$ leave-one-out (LOO) iteration, we estimated the posterior probabilities of the two programmatic action categories for the left out EU and compared the probabilities with expert consensus classification (Supplementary Table 2).

For each of the unclassified EUs ($n = 14$), we calculated the empirical probability that each of its posterior SCR values fell below an example threshold, computed as the proportion of the posterior SCR distribution below the threshold.

## Sensitivity analyses

We conducted a series of sensitivity analyses that varied age ranges, single- vs dual-antigen testing, and SCR model complexity. We estimated seroprevalence and SCR in the age ranges 1–3 and 1–9 years to determine if estimates were sensitive to the age range included and compared IgG antibody responses to dual antigens (Pgp3 + CT694) versus single antigen (Pgp3). We compared SCR estimation with or without the assumption of seroreversion, which we assumed to be 6 per 100 child-years for Pgp3, near the upper range of estimates from longitudinal studies in near-elimination and endemic settings[32–35]. A final sensitivity analysis compared SCR estimates from the Bayesian MCMC approach with a simplified approach that estimated the same SCR parameter ($\lambda$) within a generalized linear model using maximum likelihood and robust standard errors (details in Supplementary Information Text). An additional analysis illustrates how the approach could be used to develop SCR thresholds corresponding to interruption of ocular *C. trachomatis* transmission using more stringent definitions based on PCR prevalence data (details in Supplementary Information Text).

## Ethics and inclusion

The secondary analysis protocol was reviewed and approved by the Institutional Review Board at the University of California, San Francisco (Protocol #20-33198). All primary data that contributed to the analysis were collected after obtaining informed consent from all participants or their guardians under separate, local human subjects research protocols in accordance with the Declaration of Helsinki. Members from each contributing primary research study have participated as collaborators and co-authors on the present analyses from their initial stages, including the design, interpretation, and summary of results. Co-authors were nominated by each study's principal investigator to represent the country and study teams that originally contributed the data. De-identified datasets made public through this analysis have been reviewed and approved by representatives from each study and conform with ethical

guidelines set forth in the original protocols. Analyses were led by investigators at the University of California, San Francisco with guidance and input from all co-authors to incorporate local stakeholder perspectives.

### Reporting summary

Further information on research design is available in the Nature Portfolio Reporting Summary linked to this article.

## Data availability

De-identified data and replication files are available through the Open Science Framework (https://osf.io/va8uc/, https://doi.org/10.17605/OSF.IO/VA8UC and https://doi.org/10.5061/dryad.5qfttdzhx).

## Code availability

Replication code for the analyses is available through the Open Science Framework (https://osf.io/va8uc/, https://doi.org/10.17605/OSF.IO/VA8UC).

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

## Acknowledgements

This work was supported by the National Institutes of Health (NIAID R01AI158884 to B.F.A., NIGMS R35GM147702 to S.B.). A.B.K., A.W.S, and F.T. are staff members of the World Health Organization. Disclaimer: The findings and conclusions in this report are those of the author(s) and do not necessarily represent the official position of the funding agencies. The authors alone are responsible for the views expressed in this article and they do not necessarily represent the views, decisions or policies of the institutions with which they are affiliated.

## Author contributions

Following CRediT taxonomy, conceptualization (E.K. and B.F.A.), data curation (E.K., P.A.A-T., S.G., S.B., Z.A., K.A., A.A., S.A., A.M.A., M.S.A., R.L.B., R.B., E.K.C., D.C., A.D., M.I.S.D., A.-B.S.D., C.D., B.E.E., P.M.E., K.F., K.G., E.B.G., J.H., E.M.H-E., P.H., B.K., K.K., S.K., M.K., A.B.K., R.K., P.J.L., A.G.L., R.M., B.M., M.P.M., S.J.M., H.M., B.M., B.N., J.M.N., N.P., W.P., K.R.R., C.R., P.R., E.S., L.S., F.S., A.S., O.S., A.S., Z.T., F.T., E.M.T., R.T., K.T., S.K.W., K.W., T.W., D.M.W., D.Y-M., M.Y., T.Z., J.D.K., T.M.L., A.W.S., S.D.N., D.L.M., and B.F.A.), formal analysis (E.K., P.A.T., and B.F.A.), funding acquisition (S.B., A.W.S., S.D.N., D.L.M., and B.F.A.), investigation (S.G., and D.L.M.), methodology (E.K., P.A.T., S.B., M.I.S.D., E.M.H.E., S.K.W., J.D.K., T.M.L., A.W.S., S.D.N., D.L.M., and B.F.A.), project administration (B.F.A.), resources (S.G., and D.M.L.), software (E.K., P.A.T., and B.F.A.), supervision (B.F.A.), validation (E.K., and P.A.T.), visualization (E.K. and B.F.A.), Writing – Original Draft Preparation (E.K., S.G., and B.F.A.), Writing – Review & Editing (E.K., P.A.A-T., S.G., S.B., Z.A., K.A., A.A., S.A., A.M.A., M.S.A., R.L.B., R.B., E.K.C., D.C., A.D., M.I.S.D., A.-B.S.D., C.D., B.E.E., P.M.E., K.F., K.G., E.B.G., J.H., E.M.H-E., P.H., B.K., K.K., S.K., M.K., A.B.K., R.K., P.J.L., A.G.L., R.M., B.M., M.P.M., S.J.M., H.M., B.M., B.N., J.M.N., N.P., W.P., K.R.R., C.R., P.R., E.S., L.S., F.S., A.S., O.S., A.S., Z.T., F.T., E.M.T., R.T., K.T., S.K.W., K.W., T.W., D.M.W., D.Y-M., M.Y., T.Z., J.D.K., T.M.L., A.W.S., S.D.N., D.L.M., and B.F.A.).

## Competing interests

K.K.R., P.J.H., and P.M.E. are employees of, and E.M.H.E. receives salary support from, the International Trachoma Initiative, which receives an operating budget and research funds from Pfizer Inc., the manufacturers of Zithromax® (azithromycin). The other authors declare no competing interests.

## Additional information

Everlyn Kamau [1] ✉, Pearl Anne Ante-Testard [1], Sarah Gwyn [2], Seth Blumberg [1,3,4], Zeinab Abdalla[5], Kristen Aiemjoy [6,7], Abdou Amza[8], Solomon Aragie[1,9], Ahmed M. Arzika[10], Marcel S. Awoussi[11], Robin L. Bailey [12], Robert Butcher[12], E. Kelly Callahan[13], David Chaima[14], Adisu Abebe Dawed[15], Martha Idalí Saboyá Díaz [16], Abou-Bakr Sidik Domingo[11], Chris Drakeley [12], Belgesa E. Elshafie[17], Paul M. Emerson[18], Kimberly Fornace[19], Katherine Gass[20], E. Brook Goodhew[2], Jaouad Hammou[21], Emma M. Harding-Esch [12], PJ Hooper[18], Boubacar Kadri[5], Khumbo Kalua[22,23], Sarjo Kanyi[24], Mabula Kasubi[25], Amir B. Kello [26], Robert Ko[27], Patrick J. Lammie[20], Andres G. Lescano [28], Ramatou Maliki[29], Michael Peter Masika[30], Stephanie J. Migchelsen[12], Beido Nassirou[5], John M. Nesemann [1,31], Nishanth Parameswaran[2], Willie Pomat [32], Kristen K. Renneker [18], Chrissy Roberts[12], Prudence Rymil[33], Eshetu Sata[6], Laura Senyonjo[34], Fikre Seife[35], Ansumana Sillah[24], Oliver Sokana[36], Ariktha Srivathsan [1,37], Zerihun Tadesse[6], Fasihah Taleo[38], Emma Michelle Taylor[34], Rabebe Tekeraoi[39], Kwamy Togbey[11], Sheila K. West[40], Karana Wickens [2], Timothy William[41], Dionna M. Wittberg[1], Dorothy Yeboah-Manu [42], Mohammed Youbi[21], Taye Zeru[43], Jeremy D. Keenan [1,31], Thomas M. Lietman [1,3,31,37], Anthony W. Solomon [44], Scott D. Nash[13], Diana L. Martin[2] & Benjamin F. Arnold [1,3,31]

[1]F.I. Proctor Foundation, University of California San Francisco, San Francisco, CA, USA. [2]Division of Parasitic Diseases and Malaria, Centers for Disease Control and Prevention, Atlanta, GA, USA. [3]Institute for Global Health Sciences, University of California San Francisco, San Francisco, CA, USA. [4]Department of Medicine, University of California San Francisco, San Francisco, CA, USA. [5]The Carter Center, Khartoum, Sudan. [6]Department of Public Health Sciences, School of Medicine, University of California Davis CA, Davis, CA, USA. [7]Department of Microbiology and Immunology, Mahidol University Faculty of Tropical Medicine, Bangkok, Thailand. [8]Programme National de Lutte Contre la Cecité, Niamey, Niger. [9]The Carter Center, Addis Ababa, Ethiopia. [10]Centre de Recherche et d'interventions en Sante Publique, CRISP, Niamey, Niger. [11]Ministère de la Santé et de L'Hygiène Publique, Lomé, Togo. [12]London School of Hygiene & Tropical Medicine, London, UK. [13]The Carter Center, Atlanta, GA, USA. [14]Kamuzu University of Health Sciences, Lilongwe, Malawi. [15]Amhara

Regional Health Bureau, Bahir Dar, Ethiopia. [16]Department of Prevention, Control, and Elimination of Communicable Diseases, Pan American Health Organization, Washington, DC, USA. [17]Ministry of Health, Khartoum, Sudan. [18]International Trachoma Initiative, Atlanta, GA, USA. [19]Saw Swee Hock School of Public Health, National University of Singapore, Singapore, Singapore. [20]Neglected Tropical Diseases Support Center, Task Force for Global Health, Atlanta, GA, USA. [21]Direction de l'Epidemiologie et de Lutte contre les Maladies, Rabat, Morocco. [22]University of British Columbia, Vancouver, Canada. [23]Blantyre Institute for Community Outreach, Blantyre, Malawi. [24]The National Eye Health Programme, Ministry of Health, Banjul, The Gambia. [25]Muhimbili Medical Center, Dar es Salaam, Tanzania. [26]World Health Organization Regional Office for Africa, Brazzaville, Republic of Congo. [27]University of Papua New Guinea, Port Moresby, Waigani, Papua New Guinea. [28]School of Public Health and Administration, Universidad Peruana Cayetano Heredia, Lima, Peru. [29]The Carter Center, Niamey, Niger. [30]National Eye Care Coordinator, Ministry of Health, Lilongwe, Malawi. [31]Department of Ophthalmology, University of California San Francisco, San Francisco, CA, USA. [32]PNG Institute of Medical Research, Port Moresby, Waigani, Papua New Guinea. [33]Ministry of Health, Port Vila, Vanuatu. [34]Sightsavers, Haywards Heath, UK. [35]Federal Ministry of Health, Addis Ababa, Ethiopia. [36]Solomon Islands Ministry of Health and Medical Services, Honiara, Solomon Islands. [37]Department of Epidemiology and Biostatistics, University of California San Francisco, San Francisco, CA, USA. [38]World Health Organization, Port Vila, Vanuatu. [39]Ministry of Health and Medical Services, South Tarawa, Kiribati. [40]Wilmer Eye Institute, Johns Hopkins University, Baltimore, MD, USA. [41]Subang Jaya Medical Centre, Subang Jaya, Malaysia. [42]Noguchi Memorial Institute for Medical Research, Accra, University of Ghana, Ghana, Ghana. [43]Amhara Public Health Institute, Bahir Dar, Ethiopia. [44]Global Neglected Tropical Diseases Programme, World Health Organization, Geneva, Switzerland. ✉e-mail: everlyn.kamau@ucsf.edu

