## [Peer Review file · Nature Communications]

Characterizing trachoma elimination using serology

Corresponding Author: Dr Everlyn KAMAU

Version 0:

Reviewer comments:

Reviewer #1

(Remarks to the Author)

Thank you for the opportunity to review the manuscript, "Characterizing trachoma elimination using serology." This manuscript combines data from 48 serosurveys totaling more than 60k children across the world to help define the spectrum of serologic signatures related to trachoma and how these relate to the need for additional control interventions. Ten of the co-authors (experts) used serologic, clinical and PCR data from survey sites to classify each as either "action needed," "action not needed" or "unclassified." The authors estimated the seroconversion rate from each site using age-specific seroprevalence profiles in 1–5-year-olds and then combined the posterior draws from these estimates to form two (or three) empirical probability density functions, one for each setting classification type. Using these distributions the authors then demonstrate how you can take SCR estimates (including uncertainty) and estimate the probability that a surveyed setting falls in one or the other distribution for a given prior. I believe the methods in this paper are sound and this provides a great example of how we can use serologic data to not only estimate SCR/seroprevalence but also to make decisions. I only have a few suggestions/comments:

Major(ish) Comments:

- Can the authors provide the exact data on how many of the 10 expert reviewers voted for each category for each EU? It would be helpful to see the vote distribution for the unclassified EUs in relation to their predicted class probabilities (in the 2 and 3 class models) to help understand how well the model is doing for their harder cases.
- While Figure 4 is a great summary of the leave-one out analyses, it would be also useful to see the predictions for the left-out EUs (not just the unclassified ones shown in Figure 5) compared to their classification by the expert reviewers.
- Ins 172-174 - "Data from Malawi and Ethiopia were most influential based on ... but their influence was small in the regions of the SCR near higher levels of confidence." It is not clear to me what this is based on. From Figure 4, this is not obvious to me.
- The authors used the posterior draws from SCR estimates in all EUs to form the components of the mixture distribution. While I like this approach, I think it would be worthwhile for the authors to reflect on potential overfitting of the data with this approach and how this would compare to alternative parametric distributions.
- The authors suggest they can ignore seroreversion because the modal estimates are relatively similar (Figure S7). However, the posterior distributions of SCRs could indeed be quite different and it isn't clear how this would influence the final results with the mixture models.

Minor Comments:

- The open circle in Figure 4 is very hard to see. Consider other solutions for this illustration.
- It took me a while to figure out what the right-hand plots in Figure 4 were. I suggest trying to clarify this with annotations or better caption.
- The way the paper refers to the unclassified areas as 'held out areas' may give the impression that these are somehow used as a validation set. Since these are units where there is no consensus on whether action is needed, model-based predictions for these areas will not really shed light on model performance. Consider using another term for these data (simply as unclassified surveys?).
- In 332: "be difficult to identify enough 1–3-year-olds per sampled cluster to assure survey rigor" The issue isn't identifying 1-3 year olds but getting informed consent from parents.
- Ins 190-192: "Additionally, we determined the empirical probability that an EU's SCR fell below an example threshold of 2.2 per 100 person-years, with high probability that the SCR was ≤ 2.2 for most held-out EUs" What does the last clause mean?
- I suggest adding a little more detail on what you mean by empirical distributions to the methods. I wasn't sure if this was

just smoothed modes or full posterior SCRs until I looked at the source code.

(Remarks on code availability)

Thanks for sharing the repo. I scanned the code but did not try to actually reproduce analyses. I could be mistaken but I am not sure the references to the data files actually work.

Reviewer #2

(Remarks to the Author)

• What are the noteworthy results?

As highlighted by the researchers, the findings of the study suggest a data-driven guideline development based on serology as a complementary tool as we near the endgame of trachoma elimination. The study sheds light on the importance as well as approaches for using serology for elimination related threshold determination. As there is a consensus on the limitation of current measurements, this complementary indicator will be helpful for high-burden countries to refine measurement as well as measure actual progress towards elimination. The study represents a new opportunity to develop an approach for programmatic decision-making.

• Will the work be of significance to the field and related fields? How does it compare to the established literature? If the work is not original, please provide relevant references.

The manuscript builds on existing evidence from various settings and sources. The manuscript synthesizes existing evidence from 48 serosurveys in Africa, Latin America, and the Pacific Islands.

• Does the work support the conclusions and claims, or is additional evidence needed?

In addition to the limitations already acknowledged by the researchers, there are inherent challenges associated with serological tests, such as cross-reactivity (particularly in countries with co-endemicity of various diseases), temporal limitations, interpretative issues, and variability in study quality. Considering these limitations, some of the statements could potentially lead to overstated conclusions, which might mislead countries. Therefore, it would be prudent to revisit and refine some of the conclusions and claims to ensure they accurately reflect the data. Additionally, the claim of advancing the methodology may be somewhat overstated and could benefit from a more nuanced presentation.

• Are there any flaws in the data analysis, interpretation and conclusions? Do these prohibit publication or require revision?

None

• Is the methodology sound? Does the work meet the expected standards in your field?

Yes, and some of the limitations are already acknowledged.

• Is there enough detail provided in the methods for the work to be reproduced?

Yes, the authors have provided extensive sections and details in the methods, ensuring that the work can be reproduced.

(Remarks on code availability)

Reviewer #3

(Remarks to the Author)

This manuscript is well written and describes a novel approach to estimating the seroconversion rate for trachoma. The methodology is applied using a comprehensive data set, and used to estimate need for intervention in a range of evaluation units (EUs). The authors also demonstrate how the method could be applied to other diseases or data sources. I have the following specific, relatively minor, comments:

1. The abstract outlines the work that was done, but does not report any of the actual results. It would be helpful if the example thresholds were included as a way of demonstrating the application of the method.
2. Lines 61-64 within the abstract report the details of the full data set. While this is technically correct, the main results presented in the manuscript are based on a subset of this data (1-5 years). This could be explicitly stated, particularly if some results are added (as per the comment above).
3. Line 99. Suggest revising the sentence to read "Here, we combine data for IgG antibodies, ..."
4. Line 145. The authors state that the SCR estimates are "well separated between trachoma categories". I am not sure I completely agree with this statement as there is some overlap of SCR estimates between the two trachoma categories in Figure 2A. Suggest revising this statement.
5. Line 156. There is no mention in the methods how the pooled density distribution was calculated. Is the pooling done at the EU or cluster level?
6. Line 158. The authors present results for a moderately informative prior of 80% of not needing action, and continue this example through the manuscript. Some results for a 50% non-informative prior are also presented, and the methods indicate values between 50% and 80% were applied. It would be helpful if the authors explained why they selected to use the upper value of 80% as the exemplar, and also provide guidance in the discussion about how potential users of the method should select an appropriate prior probability.
7. Line 477. Suggest adding the lambda symbol after SCR so it is clear that this parameter represents the SCR. I assume the authors measured age in years, but this should be clarified.
8. Line 481. I don't think " $\lambda_a =$ " is required since the λ_a parameter is not included in any equation.
9. Line 485. It is not clear to me how λ_j is used. Is the median SCR reported in Figure 2 the median of the cluster SCRs, or are these cluster SCRs combined in a different way to estimate the EU SCR. Additional information would be

helpful.

10. Table 1. Suggest replacing "clinical signs" in the title with trachomatous inflammation (or similar) so there is a direct link between the title and the table columns. The footnote contains "Ct = ..." but this abbreviation does not appear to be used in the table.

11. Figure 2. The sample size in the title relates to all studies, not the 34 EUs included in the figure.

12. Figure 3. The title (line 728) should refer to Fig S4, not Fig S3

13. Figure 5. There seems to be a discrepancy between the title and figure for panel A. The title says "Probability of need for trachoma program intervention in held-out evaluation units". This suggests a value of 1.0 would signify an absolute need for program intervention. But the figure actually displays the probability of action not being needed. Suggest rewording the title for greater clarity.

14. Figure S1. This figure may not be needed as it simply compiles the information already reported in Figures 1 and 5.

15. Figure S2. Why is the linear regression restricted to EUs with seroprevalence $\leq 25\%$? Suggest changing "... led to coefficients..." to "...led to the model..." as the regression equation is reported, not the coefficients.

16. Figure S6. Panel A and panel B need to be referred to in the relevant place in the title.

(Remarks on code availability)

I have accessed the code via the supplied URL, but not installed the software or run the code. The README file provides links to the data and software installation instructions.

Version 1:

Reviewer comments:

Reviewer #1

(Remarks to the Author)

The revised version addresses all of my original comments/concerns. I have nothing further to add and look forward to seeing this published.

(Remarks on code availability)

Looks good.

Reviewer #3

(Remarks to the Author)

Thank you for the consideration given to the comments from the previous review. The revisions made to the manuscript adequately address my previous comments and I have nothing further to add.

(Remarks on code availability)

Response to reviewers

Reviewer #1 (Remarks to the Author):

Thank you for the opportunity to review the manuscript, “Characterizing trachoma elimination using serology.” This manuscript combines data from 48 serosurveys totaling more than 60k children across the world to help define the spectrum of serologic signatures related to trachoma and how these relate to the need for additional control interventions. Ten of the co-authors (experts) used serologic, clinical and PCR data from survey sites to classify each as either “action needed,” “action not needed” or “unclassified.” The authors estimated the seroconversion rate from each site using age-specific seroprevalence profiles in 1–5-year-olds and then combined the posterior draws from these estimates to form two (or three) empirical probability density functions, one for each setting classification type. Using these distributions the authors then demonstrate how you can take SCR estimates (including uncertainty) and estimate the probability that a surveyed setting falls in one or the other distribution for a given prior. I believe the methods in this paper are sound and this provides a great example of how we can use serologic data to not only estimate SCR/seroprevalence but also to make decisions. I only have a few suggestions/comments:

Response:

Thank you very much for your thorough review and constructive comments, which have improved the manuscript.

Major(ish) Comments:

- **Can the authors provide the exact data on how many of the 10 expert reviewers voted for each category for each EU? It would be helpful to see the vote distribution for the unclassified EUs in relation to their predicted class probabilities (in the 2 and 3 class models) to help understand how well the model is doing for their harder cases.**

Response:

*We have now provided a supplementary table with votes from all 10 expert reviewers for 34 EUs that were categorized in the final Action needed and Action not needed groups (**Supplementary Table 2**). We also provide an additional supplementary table (**Supplementary Table 3**) that has a comparison of expert votes and the predicted class probabilities (in the 2 and 3 class models) for the unclassified EUs. Also note that the Supplementary Data 1 and the dossier for the expert review process are provided in the Open Science Framework repository alongside the data and code used in this paper.*

- **While Figure 4 is a great summary of the leave-one out analyses, it would be also useful to see the predictions for the left-out EUs (not just the unclassified ones shown in Figure 5) compared to their classification by the expert reviewers.**

Response:

*We have now provided a supplementary table (**Supplementary Table 2**) with the comparisons of expert votes and the category predictions of each left out EU.*

Additionally, we added the following text to the Methods to describe this comparison:

“In each jackknife $n-1$ leave-one-out (LOO) iteration, we estimated the posterior probabilities of the two programmatic action categories for the left-out EU and compared the probabilities with expert consensus classification (Supplementary Table 2).”

• **Ins 172-174 - “Data from Malawi and Ethiopia were most influential based on ... but their influence was small in the regions of the SCR near higher levels of confidence.” It is not clear to me what this is based on. From Figure 4, this is not obvious to me.**

Response:

*This statement has been clarified with reference to Figure 4. In the revision the text now reads: “Data from Malawi and Ethiopia were most influential based on jackknife posterior probability functions, but their influence was small in regions of the SCR near higher levels of confidence. That is, at $\geq 80\%$ probability that no further action was needed, the difference between SCR when including all data and excluding either Malawi or Ethiopia was < 0.6 . For the analysis excluding individual EU as in **Fig 4**, we show comparison of expert votes with the predicted category probabilities for each EU in **Supplementary Table 3**.”*

• **The authors used the posterior draws from SCR estimates in all EUs to form the components of the mixture distribution. While I like this approach, I think it would be worthwhile for the authors to reflect on potential overfitting of the data with this approach and how this would compare to alternative parametric distributions.**

Response:

This is a good suggestion. In response to this possibility, we completed an additional analysis to rule out the potential for over-fitting. We identified the best fitting parametric distributions of the pooled or combined SCR distributions for the two categories (likelihoods in the Bayesian mixture model). To create parametric likelihoods to use in the mixture models, we generated vectors of random data from the ‘best’ fitting distributions, e.g., with `rgamma()` or `rbeta()` and re-ran the analysis to estimate SCR thresholds. The parametric distributions fitted to SCR data are shown below:

Fig - Gamma parametric distribution fit for the Action not needed category: Fit of gamma distribution to the combined SCR estimates of the Action not needed group. The shape and rate parameters (as in the top left panel) are 1.6 and 172.3, respectively.

Fig - Beta parametric distribution fit for the Action needed category: Fit of beta distribution to the combined SCR estimates of the Action needed group. The shape parameters (as in the top left panel) are 3.2 and 31.6, respectively.

Below, we show a figure similar to Fig 3 in the manuscript where the likelihoods were derived from parametric distributions above. We concluded that the estimated SCR values that correspond to 90% posterior probability of Action not needed / Action needed are extremely similar to our current analysis (in panel A, the values are 2.3 and 4.5 in this analysis, versus 2.2 and 4.5 in the primary analysis). Given this similarity and to avoid additional moving pieces in the analysis and exposition, we decided to leave the analysis and results as they were.

- The authors suggest they can ignore seroreversion because the modal estimates are relatively similar (Figure S7). However, the posterior distributions of SCRs could indeed be quite different and it isn't clear how this would influence the final results with the mixture models.

Response:

In response to this suggestion, we repeated the primary analysis, but we estimated each EU's SCR using a reversible catalytic model that allowed for seroreversion. For each EU, the seroreversion rate was an estimated parameter with a prior exponential(1) meaning the mean duration of seropositivity of 1 year. Allowing for seroreversion in the model shifts the posterior functions to higher SCR values resulting in slightly higher thresholds (Figure below). That is, for example, the SCR values at 90% posterior probability become 2.5 (up from 2.2) for Action not needed and 5.3 (up from 4.5) for Action needed (with a prior of 80% and 20%).

Fig - Probability of Action not needed and Action needed when estimating SCR reversible catalytic models. SCR for EU were estimated while assuming seroreversion with a mean duration of seropositivity of one year.

We agree that although the actual or exact threshold values corresponding to 90% certainty of action or no action will differ depending on whether the underlying model allows for seroreversion, these shifts were relatively small in the context of how much the SCR can vary. The key insight of this sensitivity analysis is that the models used to develop guidelines for decision making must match those that will be used to analyze future population surveys.

Since the primary objective of this analysis is to provide guidance for actual public health programs, we favor a simpler modeling approach that would be more easily implemented by future program monitoring teams.

In response to this additional analysis, we have added the following clarification to the Discussion:

“Third, after we found a more complex model with seroreversion led to relatively small increases in SCR estimates (Fig S7), we assumed a simple model to estimate the SCR that ignored seroreversion. Models that ignore seroreversion are easier to fit using standard regression techniques, but it means that estimates of SCR in future surveys would need to use the same underlying model and age range if comparing estimates against thresholds developed here.”

Minor Comments:

- **The open circle in Figure 4 is very hard to see. Consider other solutions for this illustration.**

Response:

We have enlarged the open circle. However, since this figure is a sensitivity analysis and we need to show the congruence or agreement of the posterior SCR value using all data versus excluding some EUs or countries, we avoid jittering points and so it might still be difficult to see where there is a complete overlap of points.

- **It took me a while to figure out what the right-hand plots in Figure 4 were. I suggest trying to clarify this with annotations or better caption.**

Response:

The annotations and caption for Figure 4 have been revised for clarification.

The revised caption is now:

“Fig 4. Sensitivity analysis of exclusion of evaluation unit- and country-level data. A jackknife $n-1$ resampling approach was used to iteratively alter group composition in a Bayesian Mixture model (Methods). Posterior probability of No Action Needed for trachoma control removing each of 34 evaluation units (**A**) and 10 countries (**B**), in turn. All estimates assumed an 80% prior probability of no action needed. The dark lines and circles in each panel show results of the resampling approach, while the red line and circles show results using the original full dataset ($N=34$ EUs and $N=10$ countries). The circles with a ‘x’ symbol in the right of **A** and **B** indicate the mean SCR over the n jackknife subsamples: mean of the SCR values represented by the dark circles. The two most influential held-out units are labeled in each sensitivity analysis. Circles in the right of A and B mark the SCR that corresponds with specific posterior probabilities between 0.5 to 1 of No Action Needed. Overall, there was minimal effect of removing data at EU- or country-level in the higher posterior probabilities (>0.8) – our primary focus. More so, there was an overlap of posterior probabilities and corresponding SCR values of the reduced datasets with that of the original full sample”

- **The way the paper refers to the unclassified areas as ‘held out areas’ may give the impression that these are somehow used as a validation set. Since these are units where there is no consensus on whether action is needed, model-based predictions for these areas will not really shed light on model performance. Consider using another term for these data (simply as unclassified surveys?).**

Response:

We have corrected the text and keep it simply as “unclassified surveys”.

- **In 332: "be difficult to identify enough 1–3-year-olds per sampled cluster to assure survey rigor" The issue isn't identifying 1-3 year olds but getting informed consent from parents.**

Response:

We think it could be both an issue of insufficient number and difficulty getting informed consent for that age group. Both perspectives apply.

- **Ins 190-192: "Additionally, we determined the empirical probability that an EU's SCR fell below an example threshold of 2.2 per 100 person-years, with high probability that the SCR was ≤ 2.2 for most held-out EUs" What does the last clause mean?**

Response:

We have clarified this by removing the last clause "with high probability that the SCR was ≤ 2.2 for most held-out EUs". We originally meant that for most unclassified EUs there was a high probability that the posterior SCR distribution was ≤ 2.2 .

- **I suggest adding a little more detail on what you mean by empirical distributions to the methods. I wasn't sure if this was just smoothed modes or full posterior SCRs until I looked at the source code.**

Response:

We rephrase this to mean full posterior SCRs and remove the word 'empirical'.

Reviewer #1 (Remarks on code availability):

Thanks for sharing the repo. I scanned the code but did not try to actually reproduce analyses. I could be mistaken but I am not sure the references to the data files actually work.

Response:

We have confirmed that the data and R analysis files are available in the online repository and openly accessible to all. The stable URL is found in our Data and materials availability statement:

"De-identified data and replication files required to conduct the analyses are available through the Open Science Framework (<https://osf.io/va8uc/>)."

Reviewer #2 (Remarks to the Author):

- **What are the noteworthy results?**

As highlighted by the researchers, the findings of the study suggest a data-driven guideline development based on serology as a complementary tool as we near the endgame of trachoma elimination. The study sheds light on the importance as well as approaches for using serology for elimination related threshold determination. As there is a consensus on the limitation of current measurements, this complementary indicator will be helpful for high-burden countries to refine measurement as well as measure actual progress towards elimination. The study represents a new opportunity to develop an approach for programmatic decision-making.

- **Will the work be of significance to the field and related fields? How does it compare to the established literature? If the work is not original, please provide relevant references.**

The manuscript builds on existing evidence from various settings and sources. The manuscript synthesizes existing evidence from 48 serosurveys in Africa, Latin America, and the Pacific Islands.

- **Does the work support the conclusions and claims, or is additional evidence needed?**

In addition to the limitations already acknowledged by the researchers, there are inherent challenges associated with serological tests, such as cross-reactivity (particularly in countries

with co-endemicity of various diseases), temporal limitations, interpretative issues, and variability in study quality. Considering these limitations, some of the statements could potentially lead to overstated conclusions, which might mislead countries. Therefore, it would be prudent to revisit and refine some of the conclusions and claims to ensure they accurately reflect the data.

Response:

Thank you for this input. We have replied to specific critiques below, though we were unsure what the reviewer meant when referring to interpretive issues and overstated conclusions because they did not provide references or specific examples that we could work from.

- I. Cross-reactivity can affect accurate inference of seroconversion in many settings (e.g., flaviviruses), but prior research (summarized in Martin et al., 2020, PMID: 32970672 and Woodhall et al., 2018, PMID: 29983342) has demonstrated that anti-Pgp3 antibody responses are very specific to Chlamydia trachomatis infections and are very unlikely due to infection with other pathogens. We acknowledge there could be neonatal and childhood exposure to urogenital Chlamydia trachomatis which may be of concern and would contribute to the observed seroprevalence and non-zero seroconversion rate estimates even in populations that have achieved elimination as a public health problem. However, as discussed previously (Martin et al., 2020 PMID: 32970672), high seroprevalence in older children in populations with active and ongoing trachoma transmission imply antibody responses due to previous ocular Ct exposure rather than exposure due to urogenital Ct. This additional source of potential exposure for young children is incorporated in the SCR estimates and posterior distributions of the “No Action” group.*
- II. It is true variability in study quality could limit output from serological tests. However, the majority of the studies or prevalence surveys used in this analysis were conducted using very rigorous Tropical Data standardized methodologies that conform to the World Health Organization population-based prevalence surveys for trachoma. Those that were not done through Tropical data were academic or university-based randomised control trials (RCTs) drawn with probability samples and assessed or monitored for quality, as should be for all RCTs.*
- III. As for temporal limitations, in our discussion we described how assessment with repeated surveys, especially in populations in the Action not needed category, would be a useful next step. This is the relevant text “Three, we lacked sufficient data to study whether repeated surveys in a single EU provide opportunities to assess the predictive value of posterior probability estimates. Several populations were measured repeatedly over time — Wag Hemra and Woreta Town in Ethiopia and Dosso, Niger. (In Dosso, Niger, all data were combined into a single estimate, Niger-MORDOR/Dosso, but seroprevalence was $\leq 0.6\%$ in every survey). The consistently low SCRs in Woreta Town and Dosso provide some proof of concept for the approach, but repeated surveys separated by multiple years in locations with moderate probability of no action needed would lead to higher posterior probabilities of no action needed or, potentially, detect recrudescence.”*

Additionally, the claim of advancing the methodology may be somewhat overstated and could benefit from a more nuanced presentation.

Response:

When developing the approach, we conducted an informal survey of the literature and surveyed our collaborator network to find examples of data-driven guidelines for elimination in the context

of neglected tropical diseases and others such as malaria. Examples were scarce, as most guidelines seem to be based on reasonable, but qualitative judgements from expert committees. The only examples we have found of a data-driven approach was the use of existing thresholds for schistosomiasis high intensity infections to inform guidelines for microhematuria, a morbidity indicator (Wiegand et al. 2021, PMID: 34115760), and previous contributions from members of our team related to trachoma (summarized in detail in the Discussion, namely Pinsent et al 2018 PMID: 30575720, and Tedijanto et al. 2024 PMID: 37277341). Thus, we stand by the novelty of the approach and feel that paragraphs 3, 4, and 5 of the Discussion provide a nuanced interpretation of this new method vis-a-vis alternative and previous methods. We are not aware of another disease that has developed public health guidelines using serology and expect that this method could generalize to other diseases.

• **Are there any flaws in the data analysis, interpretation and conclusions? Do these prohibit publication or require revision?**

None

• **Is the methodology sound? Does the work meet the expected standards in your field?**

Yes, and some of the limitations are already acknowledged.

• **Is there enough detail provided in the methods for the work to be reproduced?**

Yes, the authors have provided extensive sections and details in the methods, ensuring that the work can be reproduced.

Response:

Thank you for your review and comments.

Reviewer #3 (Remarks to the Author):

This manuscript is well written and describes a novel approach to estimating the seroconversion rate for trachoma. The methodology is applied using a comprehensive data set, and used to estimate need for intervention in a range of evaluation units (EUs). The authors also demonstrate how the method could be applied to other diseases or data sources. I have the following specific, relatively minor, comments:

1. The abstract outlines the work that was done, but does not report any of the actual results. It would be helpful if the example thresholds were included as a way of demonstrating the application of the method.

Response:

Thank you for this suggested improvement. We revised the Abstract to highlight the example thresholds to help demonstrate application of the method and discuss implications for programs.

The revised abstract is below:

“Trachoma is targeted for global elimination as a public health problem by 2030. Measurement of IgG antibodies in children is being considered for surveillance and programmatic decision-making. There are currently no programmatic guidelines based on serology, which represents a generalizable problem in seroepidemiology and disease elimination. Here, we collate *Chlamydia*

trachomatis Pgp3 and CT694 IgG measurements from 48 serosurveys across Africa, Latin America, and the Pacific Islands (32,926 children ages 1–5 years) and propose a novel approach to estimate the probability that population *C. trachomatis* transmission is below or above levels requiring ongoing programmatic action. We determine that trachoma programs could halt control measures with >90% certainty when seroconversion rates (SCRs) are ≤ 2.2 per 100 person-years. Conversely, SCRs ≥ 4.5 per 100 person-years correspond with >90% certainty that further control interventions are needed. More extreme SCR thresholds correspond with higher levels of confidence of elimination (lower SCR) or ongoing action needed (higher SCR). This study demonstrates a robust approach for using trachoma serosurveys to guide elimination program decisions.”

2. Lines 61-64 within the abstract report the details of the full data set. While this is technically correct, the main results presented in the manuscript are based on a subset of this data (1-5 years). This could be explicitly stated, particularly if some results are added (as per the comment above).

Response:

We've added results in the abstract and highlighted the sample size used from 1-5y old children, which were those used in the primary analysis as the reviewer notes (children ages 6-9y only contributed to sensitivity analyses).

3. Line 99. Suggest revising the sentence to read "Here, we combine data for IgG antibodies, ..."

Response:

This has been corrected / revised. Please see the full text in response to referee 3, comment 1

4. Line 145. The authors state that the SCR estimates are "well separated between trachoma categories". I am not sure I completely agree with this statement as there is some overlap of SCR estimates between the two trachoma categories in Figure 2A. Suggest revising this statement.

Response:

This has been revised to read "but mostly separated between trachoma categories for the larger proportions of the combined posterior distributions". The statement is more fitting to Fig 2B.

5. Line 156. There is no mention in the methods how the pooled density distribution was calculated. Is the pooling done at the EU or cluster level?

Response:

Pooled density distributions were done at EU-level by combining EU-level posterior SCR distributions into one dataframe for each category. That is, dataframes of posterior estimates for the different EUs were combined into one dataframe per trachoma category.

In the revision, we clarified this detail in the Methods:

"To make probability statements for each category, we used a mixture model framework applied to the combined full posterior distributions of SCR estimates"

6. Line 158. The authors present results for a moderately informative prior of 80% of not needing action, and continue this example through the manuscript. Some results for a 50% non-informative prior are also presented, and the methods indicate values between 50% and 80% were applied. It would be helpful if the authors explained why they selected to use the upper value of 80% as the exemplar, and also provide guidance in the discussion about how potential users of the method should select an appropriate prior probability.

Response:

We reasoned that in settings where programs are likely approaching elimination, there would be a high prior probability that no further action would be needed. The choice of 80% (as opposed to 75% or 90% or some other prior) was a subjective decision, with the idea that it reflects reasonably high confidence that no further action would be needed but still puts some probability (20% or a 1 in 5 chance) that some additional action could be required. In other use cases, such as baseline surveys or unusual epidemiological settings, we reasoned that there is typically no strong source of information going into the survey and so an uninformative prior would be most appropriate.

In the revision, we made the following edits to help explain this more clearly. In Table 2, we provided more clear labels for the use cases and how they used different priors. In the results, we made the rationale clearer and we cited Table 2 to help reinforce the link between use cases and prior scenarios (new text underlined).

“We present main results for two sets of priors that reflect important potential use cases (Table 2). In near elimination settings, we assumed a moderately informative prior of 80% probability that a trachoma program could halt control measures, with the rationale that during or after post-treatment validation further treatments would be unlikely in the absence of recrudescence. In this scenario, the posterior probability that an EU would require no further action exceeds 90% when the SCR is ≤ 2.2 per 100 person-years (Fig 3A). Conversely, SCR values ≥ 4.5 per 100 person-years correspond with $>90\%$ certainty that the population falls in the category of EUs in which further programmatic action is needed to control transmission. The choice of a particular threshold is ultimately a policy decision based on a specified level of confidence. More stringent (lower) SCR thresholds correspond with higher levels of confidence of elimination. For example, an SCR = 1.9 per 100 person-years corresponds with a level of confidence of 95% (Fig 3A). In population surveys without strong prior information, such as baseline surveys or investigations with unusual epidemiology, an uninformative prior (50% probability of each category) may be more appropriate, with SCR value corresponding to 90% probability of no action needed equal to 1.6 per 100 person-years, down from 2.2 with an 80% prior (Fig 3B, Supplementary Table 1).”

7. Line 477. Suggest adding the lambda symbol after SCR so it is clear that this parameter represents the SCR. I assume the authors measured age in years, but this should be clarified.

Response:

These have been clarified.

8. Line 481. I don't think " $\lambda_a =$ " is required since the λ_a parameter is not included in any equation.

Response:

"Lamba_a" was a convention used to indicate invariant SCR age, but the paper could do without it. It has been removed for clarity.

9. Line 485. It is not clear to me how lambda_j is used. Is the median SCR reported in Figure 2 the median of the cluster SCRs, or are these cluster SCRs combined in a different way to estimate the EU SCR. Additional information would be helpful.

Response:

Lamba_j is used to account for random effects or the clustering structure of the data. The cluster SCRs are combined in a hierarchical approach to estimate a shared random variable which is the EU level SCR.

10. Table 1. Suggest replacing "clinical signs" in the title with trachomatous inflammation (or similar) so there is a direct link between the title and the table columns. The footnote contains "Ct = ..." but this abbreviation does not appear to be used in the table.

Response:

These have been corrected.

11. Figure 2. The sample size in the title relates to all studies, not the 34 EUs included in the figure.

Response:

This has been corrected to the right sample size (n=32926).

12. Figure 3. The title (line 728) should refer to Fig S4, not Fig S3

Response:

This has been rectified to Supplementary Fig 4.

13. Figure 5. There seems to be a discrepancy between the title and figure for panel A. The title says "Probability of need for trachoma program intervention in held-out evaluation units". This suggests a value of 1.0 would signify an absolute need for program intervention. But the figure actually displays the probability of action not being needed. Suggest rewording the title for greater clarity.

Response:

This has been clarified and reads as follows: "Probability of need or no need for trachoma program intervention in unclassified EUs."

14. Figure S1. This figure may not be needed as it simply compiles the information already reported in Figures 1 and 5.

Response:

We believe both figures are needed for visual understanding of the different trachoma categories. The value of including the unclassified EUs along with those included in the main text figure is that it shows they fell at many points along the distribution and did not constitute a large, contiguous portion of the distribution. Had they all come from the center, then removing them

from the primary analysis that estimated class probabilities would have been problematic. We tried to emphasize this point in a concise way when referencing it in the main text:

“Unclassified surveys represented a range of seroprevalence and seroconversion rates (Fig 1, Supplementary Fig 1).”

15. Figure S2. Why is the linear regression restricted to EUs with seroprevalence $\leq 25\%$? Suggest changing "... led to coefficients..." to "...led to the model..." as the regression equation is reported, not the coefficients.

Response:

This is a good question and we added clarification to the caption to explain. We limited the regression fit to the linear range of the curve. The relationship between seroprevalence and SCR is non-linear over the full range of seroprevalence, 0–100% (demonstrated more clearly in Fig 3 of this previous analysis <https://www.nature.com/articles/s41467-023-38940-5/figures/3>). Since the goal was to provide a clear relationship over values most likely to be encountered in current monitoring settings, we focused on the bulk of the EUs (41/48) that fell along the linear range of the curve. We clarified this in the revised caption:

“Supplementary Fig 2. Relationship between the seroconversion rate (SCR) and seroprevalence at the evaluation unit (EU) level among 1–5-year-olds. EUs are colored by transmission categories used in the main analysis (N=48 EUs). The right panel provides a zoomed view of estimates for populations approaching elimination, illustrating a highly linear relationship between serological summary measures. A linear regression fit to the 41 EUs with seroprevalence $\leq 25\%$ (the linear range of the curve) led to the model with intercept and slope: $SCR \sim 0 + 0.363 \times \text{Seroprev}$ ($R^2=0.99$), with SCR in units per 100 and seroprevalence in percentage points.”

16. Figure S6. Panel A and panel B need to be referred to in the relevant place in the title.

Response:

This has been revised in the figure caption. Tags for ‘A’ and ‘B’ are now in the figure caption.

Reviewer #3 (Remarks on code availability):

I have accessed the code via the supplied URL but not installed the software or run the code. The README file provides links to the data and software installation instructions.

Response:

Thank you for confirming that you could access the public replication files.